# Deuterium metabolic imaging phenotypes mouse glioblastoma heterogeneity through glucose turnover kinetics

Rui Vasco Simoes[1,2]*, Rafael Neto Henriques[1], Jonas L Olesen[3], Beatriz M Cardoso[1], Francisca F Fernandes[1], Mariana AV Monteiro[4], Sune N Jespersen[3], Tânia Carvalho[4], Noam Shemesh[1]

[1]Preclinical MRI, Champalimaud Research, Champalimaud Foundation, Lisbon, Portugal; [2]Neuroengineering and Computational Neuroscience, Institute for Research and Innovation in Health (i3S), Porto, Portugal; [3]Center of Functionally Integrative Neuroscience (CFIN) and MINDLab, Department of Clinical Medicine, Aarhus University, Aarhus, Denmark; Department of Physics and Astronomy, Aarhus University, Aarhus, Denmark; [4]Histopathology Platform, Champalimaud Research, Champalimaud Foundation, Lisbon, Portugal

## eLife Assessment

This study provides a **valuable** approach to image and analyze in vivo metabolic flux through glucose turnover kinetics in glioblastoma tumor microenvironments. The evidence for the method's validity is **convincing**, which establishes the dynamic Deuterium Metabolic Imaging technique as an effective tool enabling non-invasive exploration of various tumors.

*For correspondence:
rui.vps@gmail.com

Competing interest: The authors declare that no competing interests exist.

**Abstract** Glioblastomas are aggressive brain tumors with dismal prognosis. One of the main bottlenecks for developing more effective therapies for glioblastoma stems from their histologic and molecular heterogeneity, leading to distinct tumor microenvironments and disease phenotypes. Effectively characterizing these features would improve the clinical management of glioblastoma. Glucose flux rates through glycolysis and mitochondrial oxidation have been recently shown to quantitatively depict glioblastoma proliferation in mouse models (GL261 and CT2A tumors) using dynamic glucose-enhanced (DGE) deuterium spectroscopy. However, the spatial features of tumor microenvironment phenotypes remain hitherto unresolved. Here, we develop a DGE Deuterium Metabolic Imaging (DMI) approach for profiling tumor microenvironments through glucose conversion kinetics. Using a multimodal combination of tumor mouse models, novel strategies for spectroscopic imaging and noise attenuation, and histopathological correlations, we show that tumor lactate turnover mirrors phenotype differences between GL261 and CT2A mouse glioblastoma, whereas recycling of the peritumoral glutamate-glutamine pool is a potential marker of invasion capacity in pooled cohorts, linked to secondary brain lesions. These findings were validated by histopathological characterization of each tumor, including cell density and proliferation, peritumoral invasion and distant migration, and immune cell infiltration. Our study bodes well for precision neuro-oncology, highlighting the importance of mapping glucose flux rates to better understand the metabolic heterogeneity of glioblastoma and its links to disease phenotypes.

## Introduction

Glioblastoma (glioma grade 4 or GBM) is the most aggressive primary brain tumor in adults. The dismal prognosis of such heterogeneous tumors is mostly attributed to recurrence, associated with limited response to treatment and an infiltrative pattern that prevents full surgical resection (*Wen and Kesari, 2008*). Glioblastoma heterogeneity is reflected in the tumor microenvironment, where glioma cells constantly adapt to their evolving microhabitats, with different biophysical characteristics, progression stages, and therapy resistance (*Gillies et al., 2012*). To sustain active proliferation, cancer cells exchange metabolic intermediates with their microenvironment (*Icard et al., 2014*) and undergo metabolic reprogramming (*Pavlova and Thompson, 2016*), relying heavily on aerobic glycolysis – upregulation of glucose uptake concomitant with lactate synthesis, leading to acidification of the tumor microenvironment. While this so-called Warburg effect (*Warburg, 1956*) favors e.g. invasion (*Gatenby and Gillies, 2004*), metabolic plasticity (*DeNicola and Cantley, 2015*; *Lu et al., 2010*) is becoming increasingly associated with malignant phenotypes (*Lehuédé et al., 2016*). Namely, mitochondrial oxidation (e.g. glucose metabolism through the tricarboxylic acid cycle, TCA) is linked with microenvironment adaptation and tumor progression (*Faubert et al., 2020*).

The ability to use both glycolysis and mitochondrial oxidation pathways is a critical feature of GBM, which has been demonstrated from preclinical models to patients (*Mashimo et al., 2014*; *Tardito et al., 2015*; *Maher, 2012*). More recently, specific dependencies/proclivities towards those metabolic pathways are beginning to reveal GBM subtypes with prognostic value in human cell lines and patient-derived cells (*Immanuel et al., 2021*; *Garofano et al., 2021*; *Duraj et al., 2021*). Importantly, the latest WHO classification of central nervous system tumors now distinguishes two metabolic phenotypes of adult GBM based on the molecular assessment of a specific TCA cycle mutation (isocitrate dehydrogenase, IDH), namely into grade 2-4 gliomas (IDH-mut) and grade 4 GBM (IDH-wt) (*Park et al., 2023*). The prognostic value of GBM metabolic phenotypes clearly calls for non-invasive imaging methodologies capable of resolving the different subtypes, both for diagnosis and for treatment response monitoring. However, such methods are scarce.

Deuterium metabolic imaging (DMI) has been proposed for mapping active metabolism de novo in several tumor models (*De Feyter et al., 2018*; *Kreis et al., 2020*; *Hesse et al., 2021*; *Ip et al., 2023*; *Liu et al., 2023*; *Batsios et al., 2022*; *Montrazi et al., 2023*). While this has also been demonstrated in GBM patients, with an extensive rationale of the technique and its clinical translation (*De Feyter et al., 2018*), and more recently in mouse models of patient-derived GBM subtypes (*Low et al., 2024*), mapping glucose metabolic fluxes remains unaddressed in these tumors due to the poor temporal resolution of DMI; particularly for glucose mitochondrial oxidation. Leveraging the benefits and risks of denoising methods for MR spectroscopy (*Goryawala et al., 2020*; *Clarke and Chiew, 2022*; *Dziadosz et al., 2023*), we recently combined Deuterium Magnetic Resonance Spectroscopy ($^2$H-MRS) (*Lu et al., 2017*) with Marcheku-Pastur Principal Component Analysis (MP-PCA) denoising (*Veraart et al., 2016*) to propose Dynamic Glucose-Enhanced (DGE) $^2$H-MRS (*Simões et al., 2022*), demonstrating its ability to quantify glucose fluxes through glycolysis and mitochondrial oxidation pathways in vivo in mouse GBM, which in turn revealed their proliferation status.

Here, we develop and apply a novel rapid DGE-DMI method to spatially resolve glucose metabolic flux rates in mouse GBM and reach a temporal resolution compatible with its kinetic modeling. For this, we adapt two advances of PCA denoising – tensor MPPCA (*Olesen et al., 2023*; *Christensen et al., 2023*) and threshold PCA denoising (*Henriques et al., 2023*) – and apply them to the regional metabolic assessment of mouse GBM. First, we validated our novel approach in vivo for its ability to map glucose fluxes through glycolysis and mitochondrial oxidation in mouse GBM. Then, we investigate the potential of our new approach for depicting histopathologic differences in two mouse models of glioblastoma, including microglia/macrophage infiltration, tumor cell proliferation, peritumoral invasion, and migration. For this we used the same allograft mouse models of GBM, induced with CT2A and GL261 cell lines (*Zagzag et al., 2000*; *Seligman and Shear, 1939*; *Oh et al., 2014*; *Seyfried et al., 1992*; *Martínez-Murillo and Martínez, 2007*), but at more advanced stages of progression (*Simões et al., 2022*). Since DMI is already performed in humans, including in glioblastoma patients (*De Feyter et al., 2018*), DGE-DMI could be relevant to improve the metabolic mapping of the disease.

## Results

### MRI assessment of mouse GBM

Multi-parametric MRI provided a detailed characterization of each cohort at an endpoint. Volumetric T2-weighted MRI indicated consistent tumor sizes across CT2A and GL261 cohorts ($58.5\pm7.2$ mm$^3$). GL261 tumors were studied sooner after induction ($17\pm0$ vs $30\pm5$ d post-injection, p=0.032), explaining the lower animal weights in this cohort ($22.4\pm0.6$ vs $25.7\pm0.9$ g, p=0.017). DCE T1-weighted MRI indicated higher vascular permeability ($0.85\pm0.11$ vs $0.43\pm0.05$ $\cdot10^{-2}$/min, p=0.012) and a tendency for larger extracellular volume fractions ($0.26\pm0.03$ vs $0.18\pm0.02$, p=0.056) in the GL261 tumors compared to CT2A. However, DCE T1-weighted MRI was carried out only in 80% of the mice due to time restrictions. This information is detailed in (*Supplementary file 1a, table 1*), where the quantitative assessment of DGE-DMI, DCE-T1, and histologic parameters is displayed for tumor and peritumor border regions (P-Margin), based on ROI analysis.

### DGE-DMI in mouse GBM

Tumor metabolic assessment was performed with DGE-DMI in CT2A vs GL261 cohorts. No differences in RF coil quality or magnetic field homogeneity were detectable between the two cohorts: Q-factor $^2$H, $175\pm8$ vs $176\pm9$ (p=0.8996), respectively; FWHM $^1$H (VOI), $29.2\pm6.6$ vs $26.0\pm4.3$ Hz (p=0.3837), respectively. DGE-DMI was used to map the natural abundance of semi-heavy water signal (DHO) as well as the dynamic conversion of deuterium-labeled glucose (Glc) to its downstream products, lactate (Lac) and glutamate-glutamine (Glx) pools, in tumor and peritumor brain regions (*Figure 1A*). Tensor PCA denoising improved the spectral quality compared to the original data, without any depictable effects in the relative spatial distributions of signal-to-noise-ratio (SNR, *Figure 1—figure supplement 1*), leading to a consistent and significant ~threefold SNR increase across all the subjects (from $6.4\pm0.1$ before denoising to $20.1\pm0.4$ after denoising, *Supplementary file 1a, table 1*).

Spectral quantification of DGE-DMI data in each voxel and time point rendered time-course de novo concentration maps for each metabolite (DHO, Glc, Glx, and Lac), in both GBM cohorts (*Figure 1B*). Voxel-wise averaging of DGE-DMI time-course data after Glc injection generated average metabolic concentration maps for each tumor (*Figure 1C*). Thus, Lac concentration was visually higher in the tumor regions, due to enhanced glycolysis; whereas Glx was more apparent in the adjacent non/peritumoral areas, consistent with a more prevalent oxidative metabolism in the normal brain. Kinetic fitting of DGE-DMI time-course concentration maps rendered glucose flux maps, namely its maximum consumption rate ($V_{max}$) and flux rates through glycolysis ($V_{lac}$ and $k_{lac}$) and mitochondrial oxidation ($V_{glx}$ and $k_{glx}$) (*Figure 1D*). Both cohorts displayed higher glycolytic metabolism in the tumors and more pronounced glucose oxidation in non-tumor regions, aligned with average concentration maps.

### Histopathology assessment of GBM cohort differences

Histopathological analysis consisted of screening the CT2A and GL261 brain tumors for morphological features, including a qualitative assessment of cell density, hemorrhage, tumor vessels, necrosis, quantification of peripheral infiltration, and quantification of tumor proliferation index, while blinded to the in vivo MRI/MRS data – *Supplementary file 1b, table 2*. Thus, tumors were scored individually for the following stromal-vascular phenotype, as in *Simões et al., 2022*, where: pattern I corresponds to predominance of small vessels, complete endothelial cell lining, and sparse hemorrhages; pattern II to vasodilation and marked multifocal hemorrhages; pattern III to predominance of necrosis of the vascular wall, incomplete endothelial cell lining, vascular leakage, and edematous stroma; and pattern IV to tumors with absence of clear vascular spaces and edematous stroma.

Stromal-vascular phenotypes reflected the more advanced stages of tumor progression in which these tumors were collected, as compared to our previous study (*Simões et al., 2022*). Particularly, CT2A (n=5) presented patterns I to III, whereas all GL261 (n=5) matched pattern IV *Supplementary file 1a, table 1*. Moreover, the increased infiltrative and migratory characteristics of GL261 compared to CT2A tumors were evident in their irregular tumor borders and higher incidence of secondary brain lesions (*Figure 2A*). These findings collectively suggest a more invasive and aggressive pattern of GL261 tumors, characterized by reduced cell-cell adhesion and enhanced migratory potential compared to CT2A. Such phenotype differences were reflected in the regional infiltration by microglia/macrophages: significantly higher at the CT2A peritumoral margin (P-Margin) compared to GL261, and slightly higher in the tumor region as well (*Figure 2B*). Further quantitative regional

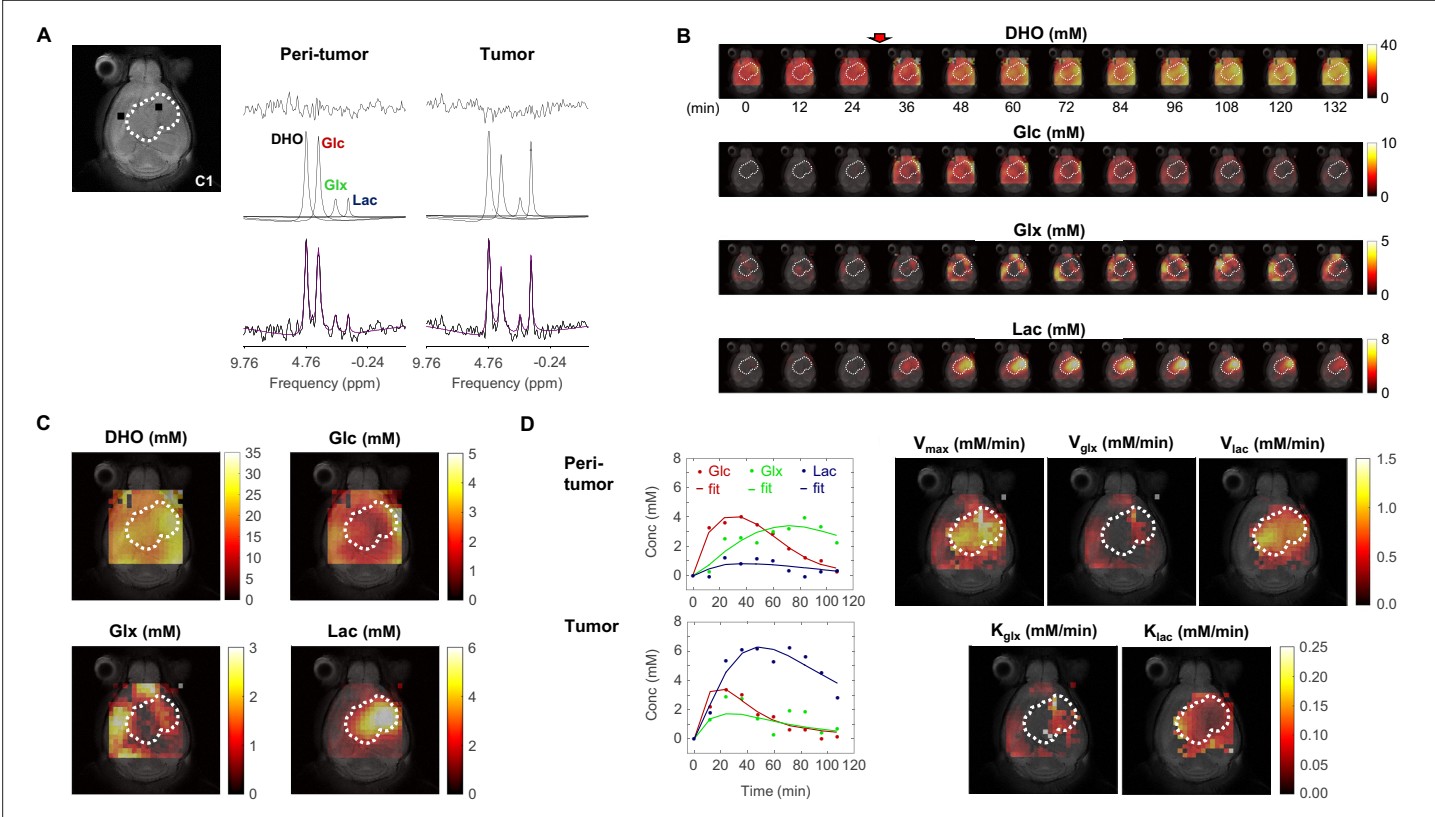

**Figure 1.** Metabolic concentration and flux maps from DGE-DMI in mouse glioblastoma (GBM). Example of a CT2A tumor (**C1**). (**A**) T2-weighted reference image (top-left) displaying the tumor region (dashed lines) and representative peritumor and tumor voxels (back dots), and respective spectral quantifications (right-side): bottom, raw spectrum (black) with overlaid estimation (purple); center, individual components for each metabolite peak (black - semi-heavy water, DHO (black); deuterated glucose, Glc (red); and glucose-derived glutamate-glutamine and lactate, Glx (green) and Lac (blue)); top, residual. (**B**) Time-course de novo concentration maps for each metabolite (mM) following Glc i.v. injection (red arrow). (**C**) Average concentration maps for each metabolite after Glc injection. (**D**) Time-course concentration plots for each metabolite (dots) and respective kinetic fitting (straight lines), displayed for the peritumor and tumor voxels shown in A (same color codes) and applied to all the voxels to generate glucose flux maps: maximum consumption rate ($V_{max}$); and respective individual rates for lactate synthesis ($V_{lac}$) and elimination ($k_{lac}$), and glutamate-glutamine synthesis ($V_{glx}$) and elimination ($k_{glx}$).

The online version of this article includes the following figure supplement(s) for figure 1:

**Figure supplement 1.** Tensor principal component analysis (PCA) denoising improves DGE-DMI SNR in mouse glioblastoma (GBM).

**Figure supplement 2.** Quantification of DGE-DMI data.

**Figure supplement 3.** Tensor principal component analysis (PCA) denoising has no overall effect on pixel distributions of DGE-DMI metabolic concentration maps in pooled glioblastoma (GBM) cohorts.

**Figure supplement 4.** Tensor principal component analysis (PCA) denoising has no effect on pixel detectability or glioblastoma (GBM) cohort differences of DGE-DMI time-course average metabolic concentration maps.

**Figure supplement 5.** Kinetic modeling of DGE-DMI time-course concentration maps data.

**Figure supplement 6.** Tensor principal component analysis (PCA) denoising increases pixel densities of DGE-DMI metabolic flux maps in pooled glioblastoma (GBM) cohorts.

**Figure supplement 7.** Tensor principal component analysis (PCA) denoising improves pixel detectability without affecting glioblastoma (GBM) cohort differences of DGE-DMI metabolic flux maps.

**Figure supplement 8.** Glucose consumption rates in mouse glioblastoma (GBM) tumors following ROI averaged raw data.

**Figure supplement 9.** Effect of changing $v_e$ on the metabolic maps derived from kinetic modeling.

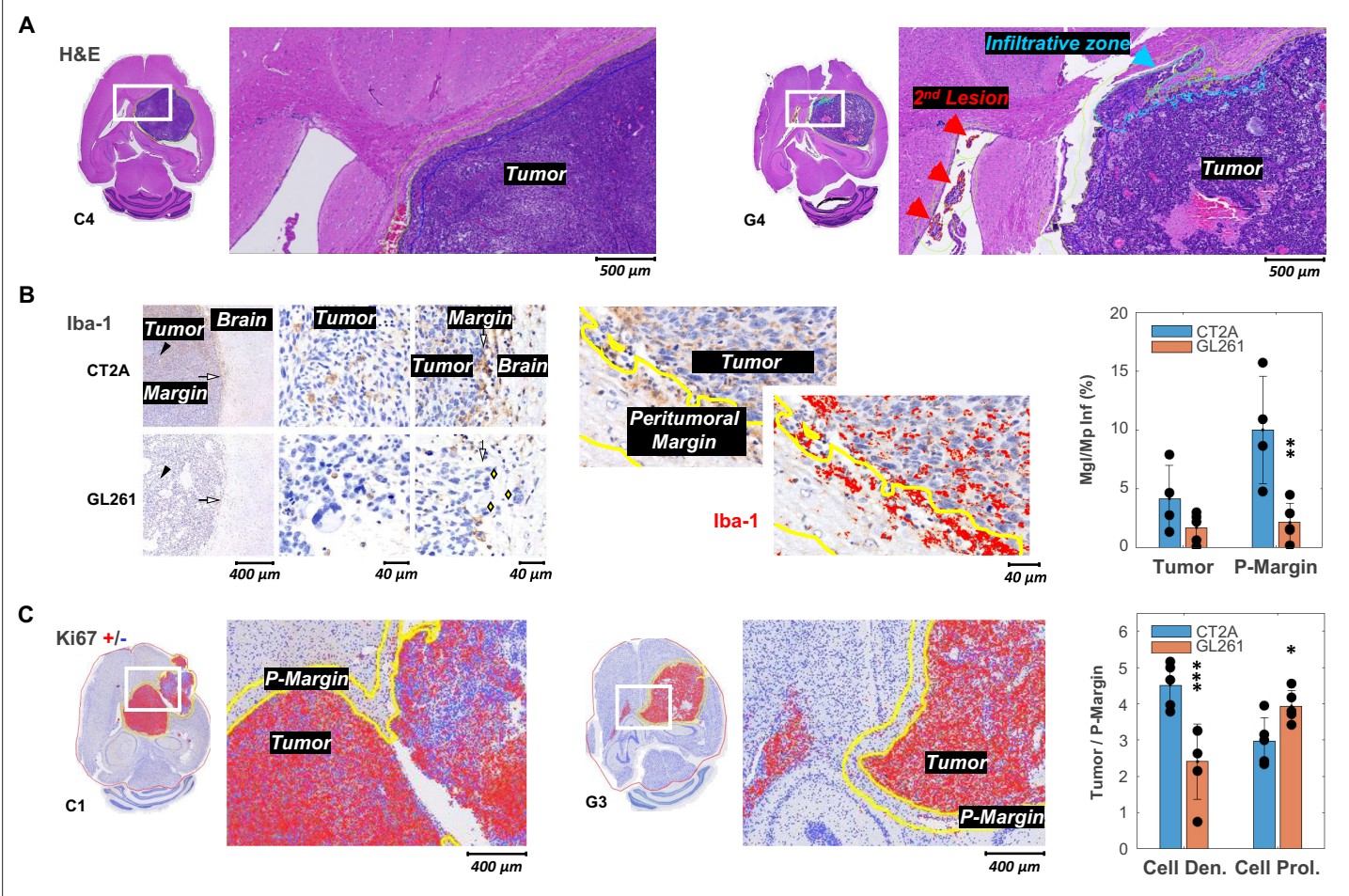

**Figure 2.** Histopathologic and immunohistochemical assessment in two mouse models of glioblastoma (GBM). (**A**) H&E-stained sections with high magnification to highlight annotations of tumor, infiltrative zones in the tumor margin (blue), and secondary lesion (red), in CT2A and GL261 tumors (subjects C4 and G4, respectively). (**B**) Iba-1 immunostained sections showing microglia/macrophage (Mgl/Mp) infiltration in CT2A and GL261 tumors: left panels, tumor core (black arrowhead) and tumor margin (white arrow) relative to the adjacent brain parenchyma; middle and right panels, depicting more infiltration by microglial/macrophage in CT2A tumors, also with clearer well-demarcated margin where IBA-1-positive cells are more densely concentrated compared to the more diffuse and irregular infiltration seen in the GL261 model; GL261 show poorly demarcated tumor border where tumor cells infiltrate the brain parenchyma (yellow diamonds); center panels, Iba-1 ROI quantification in tumor and peritumoral margin (P-Margin, yellow lines), and with red mask overlay of Iba-1 positive cells; right panel, quantification of mean Iba-1 positive area in Tumor and P-Margin regions from each cohort: GL261 (n=5) and CT2A (n=4; C2 sample excluded due to peritumoral hemorrhage/vascular ectasia, which distorted the peritumoral area and impaired proper assessment of peritumoral infiltration). (**C**) Ki67 immuno-stained sections with overlaid detection of positive (red) and negative (blue) cells; and high magnification to highlight annotations of tumor and peritumor border (P-Margin, yellow lines), in CT2A and GL261 tumors (subjects C1 and G3, respectively); and GBM cohort differences in tumor/P-Margin ratios of cell density and cell proliferation from GL261 (n=5) and CT2A (n=5) cohorts. Dots representative of average values for each subject. CT2A vs GL261: *p<0.05; **p<0.01; ***p<0.001; unpaired *t*-test. Error bars: standard deviation.

The online version of this article includes the following figure supplement(s) for figure 2:

**Figure supplement 1.** Gross inter-subject correlations of averaged metabolic maps.

analysis of Tumor-to-P-Margin ROI ratios revealed: (i) 47% lower cell density (p=0.004) and 32% higher cell proliferation (p=0.026) in GL261 compared to CT2A (***Figure 2C***, ***Supplementary file 1c, table 3***); and (ii) strong negative correlations in pooled cohorts between microglia/macrophage infiltration and cellularity (*R*=−0.91, p=<0.001) or cell density (*R*=−0.77, p=0.016), suggesting more circumscribed tumor growth with higher peripheral/peritumoral infiltration of immune cells.

Despite the more advanced stages of tumor progression, the results were largely consistent with the marked morphological differences between the two models (***Simões et al., 2022***): CT2A with dense, cohesive, and homogeneous cell populations (***Figure 2A***, left-side); GL261 displaying marked

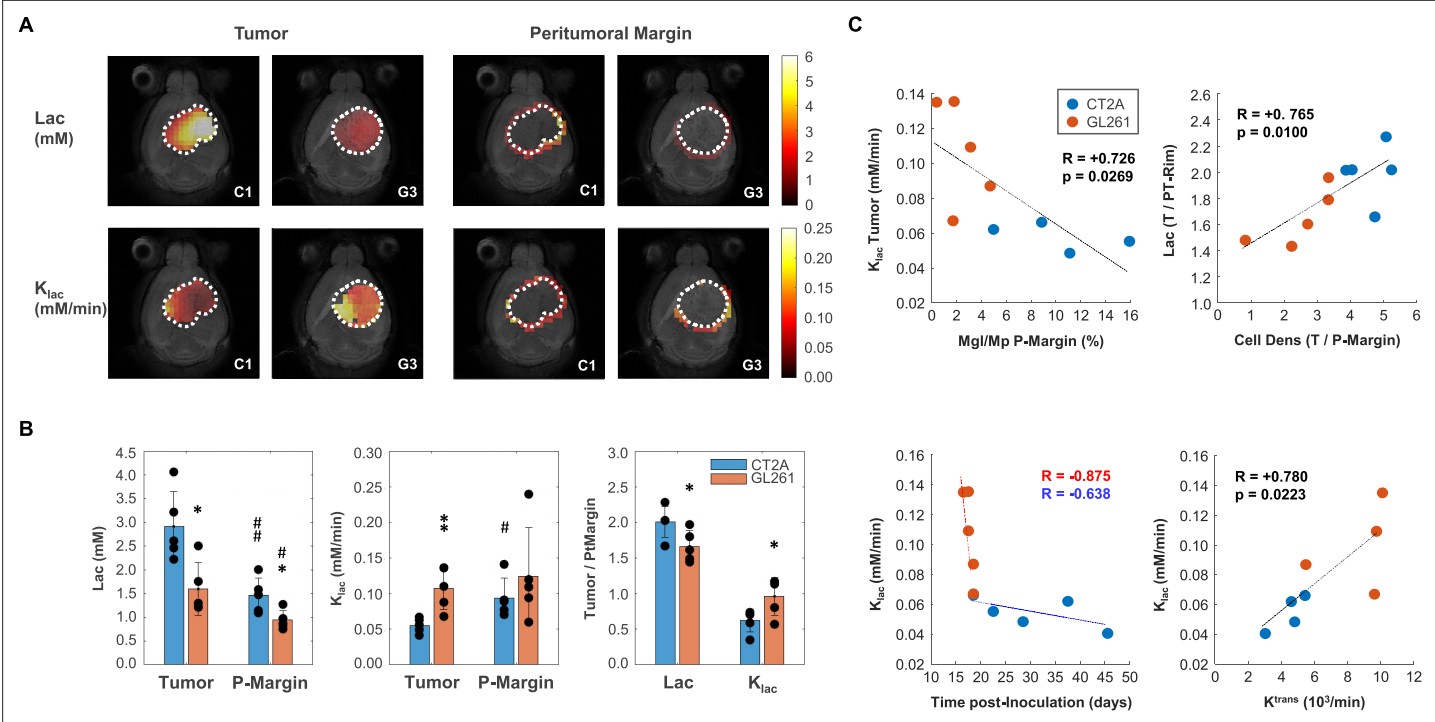

**Figure 3.** Mouse glioblastoma (GBM) models with different histopathologic phenotypes underlied by regional differences in lactate metabolism. (**A**) Metabolic maps of de novo lactate accumulation (mM) and respective consumption/elimination rates (mM/min), in tumor and peritumor border regions (P-Margin, delineated by dashed lines) of CT2A and GL261 tumors (subjects C1 and G3, respectively). (**B**) GBM cohort differences in de novo lactate accumulation (Lac) and consumption/elimination rates ($k_{lac}$). (**C**) Strong linear correlations indicated by the Person correlation coefficient, (**R**) of: top-left, Tumor lactate consumption/elimination rates with P-Margin infiltration of microglia/macrophages in pooled cohorts; top-right, Tumor-to-P-Margin ratios of lactate accumulation and cell density in pooled cohorts; bottom, lactate consumption/elimination rates with (left-side) time post-tumor inoculation in each cohort, and (right-side) tumor vascular permeability in pooled cohorts. CT2A (n=5) vs GL261 (n=5): *p<0.05; **p<0.01; unpaired *t*-test. Tumor (n=5, each cohort) vs P-Margin (n=5, each cohort): #p<0.05; ##p<0.01; paired *t*-test. Error bars: standard deviation. Bar plot dots are representative of average pixel values for each subject.

The online version of this article includes the following figure supplement(s) for figure 3:

**Figure supplement 1.** Intra-tumor pixel-wise correlations between metabolic and permeability metrics.

heterogeneity, with poorly cohesive areas and more infiltrative growth (*Figure 2A*, right-side). Quantitative assessment (nuclear counts) further confirmed a nearly twofold lower cell density of GL261 tumors compared to CT2A (4.9 vs 8.2×$10^3$ cells/μm$^2$, p<0.001) despite their similar proliferation index (*Supplementary file 1a, table 1*); and tumor cell density correlated with cell proliferation, strongly for CT2A (*R*=0.96, p=0.009) and the same tendency detected for GL261 (*R*=0.74, p=0.151).

Tumor volume and whole-brain gross assessment of cell density, cell proliferation, and glucose metabolism also revealed strong inter-subject correlations in both cohorts (*Figure 2—figure supplement 1*): de novo glutamate-glutamine accumulation decreased with tumor size (R $_{CT2A/\ GL261/\ pooled}$: –0.597/-0.753 / -0.455), consistent with its role as a marker of oxidative metabolism in the normal brain; lactate synthesis rate increased with cellularity (R $_{CT2A/\ GL261/\ pooled}$:+0.921 / +0.685 / +0.852), also aligned with enhanced glycolysis in growing tumors; whereas glucose accumulation reflected cell proliferation (R $_{CT2A/\ GL261/\ pooled}$: +0.469 / +0.528 / +0.440).

## Regional assessment of glucose metabolism in the tumor microenvironment

We then accessed regional glucose metabolism (*Figure 3*). Initial intra-tumor analysis of DGE-DMI and DCE-T1 maps (pixel-wise correlations in tumor ROIs) indicate stronger correlations between de novo lactate accumulation (Lac) and vascular permeability ($k^{trans}$) in both cohorts (R between [+0.306 + 0.741]), and extracellular space ($v_e$) to some extent (R between [–0.084+0.804]) – both less apparent without tensor PCA denoising (R between [+0.089+0.647] and [–0.160+0.684], respectively) (*Figure 3—figure*

*supplement 1*). Such accumulation of lactate according to local vascular permeability mostly reflected regional differences in glycolytic fluxes ($V_{lac}$: R between [–0.066+0.510]), rather than lactate elimination rates ($k_{lac}$: R between [–0.643+0.460]). No additional correlations were detected.

GL261 tumors accumulated significantly less lactate in the core (1.60±0.25 vs 2.91±0.33 mM: –45%, p=0.013) and peritumor margin regions (0.94±0.09 vs 1.46±0.17 mM: –36%, p=0.025) than CT2A – *Figure 3A–B*, (*Supplementary file 1a, table1*). Consistently, tumor lactate accumulation correlated with tumor cellularity in pooled cohorts (*R*=0.74, p=0.014). Then, lower tumor lactate levels were associated with higher lactate elimination rate, $k_{lac}$ (0.11±0.1 vs 0.06±0.01 mM/min: +94%, p=0.006) – *Figure 3B* – which in turn correlated inversely with peritumoral margin infiltration of microglia/macrophages in pooled cohorts (*R*=−0.73, p=0.027) - *Figure 3C*. Further analysis of Tumor/P-Margin metabolic ratios (*Supplementary file 1c, table 3*) revealed: (i) +38% glucose (p=0.002) and −17% lactate (p=0.038) concentrations, and +55% higher lactate consumption rate (p=0.040) in the GL261 cohort; and (ii) lactate ratios across those regions reflected the respective cell density ratios in pooled cohorts (*R*=0.77, p=0.010) – *Figure 3C*. Finally, lactate elimination rate correlated inversely with 'tumor age' (time post-induction) in pooled cohorts (*R*=−0.66, p=0.039), and more consistently with tumor vascular permeability ($k^{trans}$: *R*=0.78, p=0.022) (*Figure 3C*), rather than washout rate ($k_{ep}$: *R*=0.61, p=0.109).

## Association between glucose metabolism and peritumoral invasion and migration

Finally, we investigated the association between glucose metabolism and phenotypic features of tumor aggressiveness, namely cell proliferation and tumor cell invasion and migration associated with secondary brain lesions. Only the more infiltrative GL261 cohort displayed inter-subject associations between tumor cell proliferation (Ki67[+] %) and metabolism, namely inverse correlations with tumor border/peritumoral glucose oxidation rate ($V_{glx}$: *R*=−0.91, p=0.030) and glucose-derived glutamate-glutamine elimination rate ($k_{glx}$: *R*=−0.99, p<0.001). Regrouping subjects according to glioma cell invasion and migration concomitant with secondary brain lesions (presence: C1, G3, G4, G5; vs. absence: C2, C3, C4, C5, G1, G2) revealed lower de novo glutamate-glutamine levels in peritumor brain regions (Glx: –37%, p=0.013), which were associated with its higher elimination rate ($k_{glx}$: +69%, p=0.012) – *Figure 4*.

## Discussion

Glioblastomas are aggressive brain tumors with a poor prognosis, largely due to their inter- and intra-tumor heterogeneity and lack of non-invasive methods to assess it. Here, we developed and applied a DGE-DMI approach capable of generating metabolic concentration maps and flux rates in two mouse models of glioblastoma, based on unambiguous spectral quantification according to quality criteria. Our results suggest that glycolytic lactate turnover mirrors phenotype differences between the two glioblastoma models, whereas recycling of the glucose-derived glutamate-glutamine pool could underlie glioma cell migration leading to secondary lesions. This information became more readily available when using the tensor PCA method for spectral denoising.

Tensor PCA denoising increased spectral SNR by ~ threefold, consistently improving spectral quality observed in tumor and peritumoral regions without altering the spatiotemporal profiles of the metabolic concentration maps (*Figure 1—figure supplement 2*). While this had no apparent effect on metabolic concentration maps (*Figure 1—figure supplements 3–4*), it significantly improved the kinetic modeling performance (*Figure 1—figure supplement 5*) and rendered better quality metabolic flux maps in CT2A and GL261 cohorts. Thus, 63% increased pixel detectability enabled capturing more spatial features in the latter without affecting parameter estimates or introducing group differences (*Figure 1—figure supplements 6 and 7*).

Gross whole brain analysis revealed strong inter-subject correlations in both cohorts, such as higher lactate synthesis rate with increasing cellularity – consistent with enhanced glycolysis in growing tumors – whereas intra-tumor pixel-wise analysis suggested lactate accumulation according to local vascular permeability, mostly associated with regional differences in glycolytic fluxes. Such pixel-wise analyses might be misleading since de novo lactate diffuses quickly within tumor extracellular spaces and peritumoral regions (*Provent et al., 2007*), with spatiotemporal dynamics not fully captured by

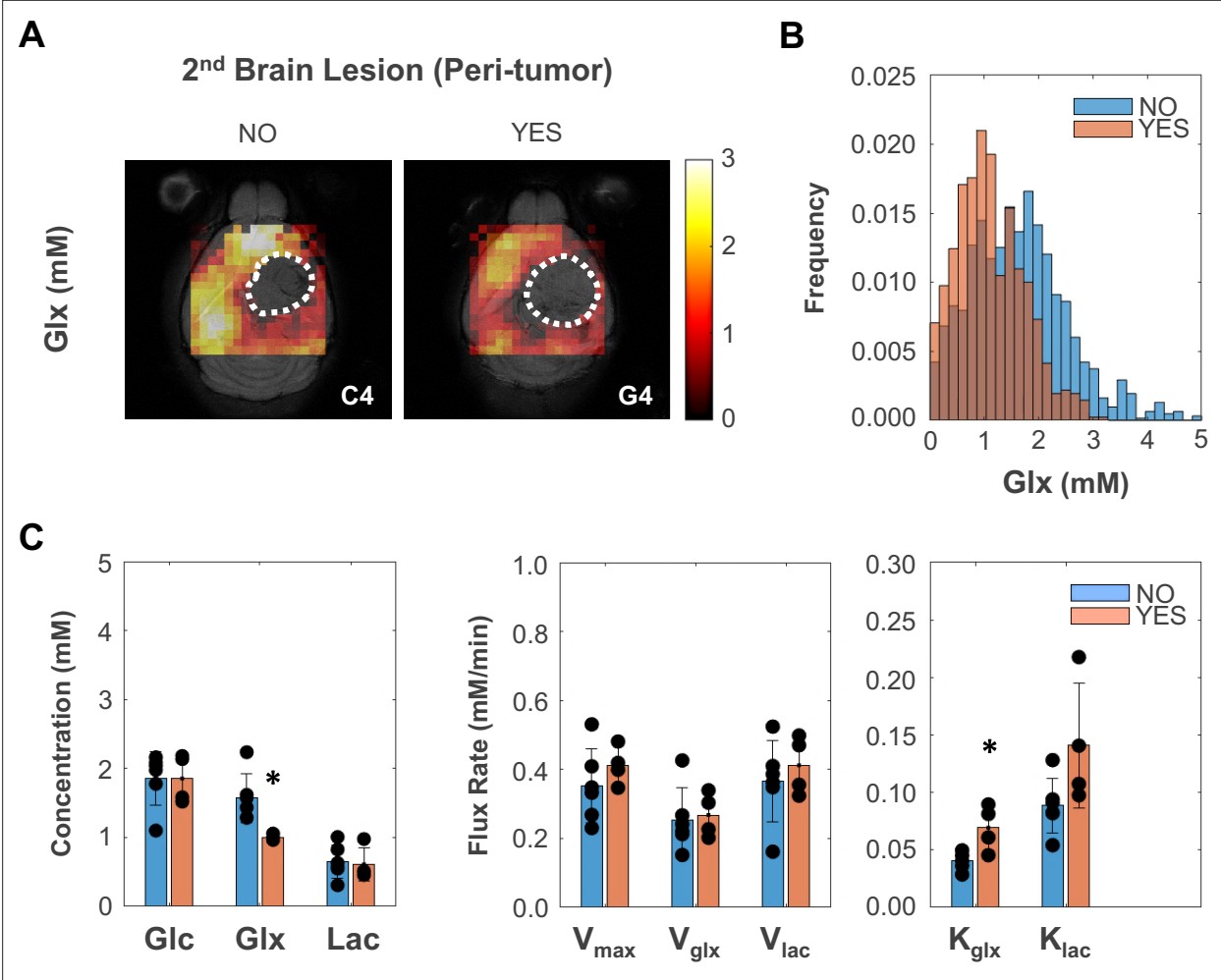

**Figure 4.** Peritumoral metabolic changes consistent with recycling of the glutamate-glutamine pool mirror glioblastoma (GBM) infiltration and migration leading to secondary brain lesions. (**A**) Metabolic maps (Glx) of peritumoral regions without and with secondary brain lesions (C4 and G4 tumors, respectively). (**B**) Histogram distributions of peritumoral Glx accumulation in pooled GL261 and CT2A cohorts displaying secondary brain lesions (n=4) vs without (n=6). (**C**) Bar plot comparison of mean values, showing significant decreases in peritumoral glutamate-glutamine accumulation (Glx) and increases in its consumption/elimination ($k_{glx}$) in pooled GL261 and CT2A cohorts displaying secondary brain lesions (n=4; vs n=6 without): *$p<0.05$; unpaired *t*-test. Bar plot dots representative of average pixel values for each subject. Error bars: standard deviation.

The online version of this article includes the following figure supplement(s) for figure 4:

**Figure supplement 1.** Tumor metabolic changes mirror glioblastoma (GBM) infiltration and migration leading to secondary brain lesions.

DGE-DMI. Namely, water diffusion in GL261 tumors in vivo (apparent diffusion coefficient ~$10^{-3}$ mm²/s *Simões et al., 2008b*; *Roberts et al., 2020*) extends beyond the in-plane voxel area (0.56×0.56 = 0.31 mm²) during each time frame (12 min). Thus, we focused instead on inter-tumor ROI analysis of glucose metabolic fluxes, in tumor and peritumoral (border) regions.

Compared to our previous study using the same GBM models (*Simões et al., 2022*), larger tumors (59±7 vs 38±3 mm3) display more disrupted stromal-vascular phenotypes (H&E scores: CT2A I-III vs I; GL261, IV vs I-IV) and weaker cell-cell interactions (lower cohesiveness) (*Supplementary file 1b, table 2*), associated with lower vascular permeability ($k^{trans}$: 6±1 vs 14±1 $10^3$ /min) and leading to lower glucose oxidation rates ($V_{glx}$: 0.28±0.06 vs 0.40±0.08 mM/min), but remarkably similar glycolytic fluxes ($V_{lac}$: 0.59±0.04 vs 0.55±0.07 mM/min). Thus, glycolysis flux rates are relatively well preserved across GL261 and CT2A mouse GBM models, regardless of tumor volume and vascular permeability.

GL261 tumors were examined earlier after induction than CT2A (17±0 vs 30±5 d, p=0.032), displaying similar volumes (57±6 vs 60±14, p=0.813) but increased vascular permeability (8.5±1.1 vs 4.3±0.5 $10^3$ /min:+98%, p=0.001), more disrupted stromal-vascular phenotypes and infiltrative

growth (5/5 vs 0/5), consistent with significantly lower tumor cell density ($4.9 \pm 0.2$ vs $8.2 \pm 0.3$ $10^{-3}$ cells/$\mu m^2$: –40%, p<0.001) and lower peritumoral rim infiltration of microglia/macrophages ($2.1 \pm 0.7$ vs $10.0 \pm 2.3\%$: –77%, p=0.008). Such GBM cohort differences were markedly reflected in their regional lactate metabolism. Thus, GL261 tumors accumulated roughly –40% less lactate in tumor and peri-tumor border regions, associated with +94% higher lactate elimination rate rather than glycolytic rate differences in tumor regions, as could be assumed solely based on metabolic concentration maps.

Tumor vs peritumor border analyses further suggest that lactate metabolism reflects regional histologic differences: lactate accumulation mirrors cell density gradients between and across the two cohorts; whereas lactate consumption/elimination rate coarsely reflects cohort differences in cell proliferation, and inversely correlates with peritumoral infiltration by microglia/macrophages across both cohorts. This is consistent with GL261's lower cell density and cohesiveness, more disrupted stromal-vascular phenotypes, and infiltrative growth pattern at the peritumor margin area, where less immune cell infiltration is detected and relatively lower cell division is expected (*Darmanis et al., 2017*). Altogether, our results suggest an increased lactate consumption rate (active recycling) in GL261 tumors with higher vascular permeability, e.g., as a metabolic substrate for oxidative metabolism (*Torrini et al., 2022*) promoting GBM cell survival and invasion (*Colen et al., 2011*), aligned with the higher respiration buffer capacity and more efficient metabolic plasticity of GL261 cells than CT2A (*Simões et al., 2022*). While, lactate shuttling within the tumor microenvironment is also reported in other tumor types, between cancer cells (*Sonveaux et al., 2008*) and between cancer and stromal cells (*Patel et al., 2017*; *Végran et al., 2011*), it should be noted that oxidative phosphorylation inefficiency has been extensively documented in cancer cells, including GBM (*Seyfried et al., 2024*), largely associated with hypoxic niches and in agreement with our measurements of lower glucose oxidation rate ($V_{glx}$) in tumor vs. peritumoral regions.

The lower glucose oxidation rates measured in this study compared with smaller, better perfused tumors (*Simões et al., 2022*), are in good agreement with our previous data indicating quick adaptation of this pathway flux according to oxygen availability in the tumor microenvironment (*Simões et al., 2022*). Under such physiological conditions – underlying more advanced progression stages, reflected in more disrupted stromal-vascular phenotypes – tumor glucose oxidation rate was not associated with cell proliferation index, consistent with previous observations (*Simões et al., 2022*). Instead, tumor cell proliferation was inversely correlated with tumor border/peritumoral glucose oxidation rate and glucose-derived glutamate-glutamine elimination rate in more infiltrative GL261 tumors; but not in CT2A. This observation is consistent to some extent with GL261 cells' and tumor's ability to modulate mitochondrial metabolism according to their microenvironment (e.g. oxygen availability *Simões et al., 2022*), which is likely to occur during their progression from more circumscribed/local cell proliferation towards more disrupted stromal-vascular phenotypes, associated with significantly lower peri-tumoral immune cell infiltration and higher tumor invasion compared to CT2A.

Notably, glucose-derived glutamate-glutamine displayed –37% lower levels and +69% higher elimination rate in peritumor regions of mouse brains bearing secondary GBM lesions (respective primary tumors displaying +146% increased glucose oxidation rate, detectable only with tensor PCA denoising – *Figure 4—figure supplement 1*). This could be associated with glutamate-glutamine-driven mitochondrial metabolism, through the TCA cycle coupled with oxidative phosphorylation (more prevalent in the normal brain) and/or via substrate-level phosphorylation for ATP synthesis – glutaminolysis (as reported in glioma cells, e.g. CT2A *Chinopoulos and Seyfried, 2018*). While patient-derived xenografts and de novo models would be more suited to recapitulate human GBM heterogeneity and infiltration features, and genetic manipulation of glycolysis and mitochondrial oxidation pathways could be relevant to ascertain DGE-DMI sensitivity for their quantification, our observations are well aligned with the pivotal role of mitochondrial metabolism in cancer cells with higher motile potential, as reported in human GBM (*Saurty-Seerunghen et al., 2022*) and in mouse and human breast cancer cell lines (*Simões et al., 2015*; *Nóbrega-Pereira et al., 2023*). Particularly, the dynamics of glutamate shuttling underlying neuronal-glioma cell communication and promoting GBM infiltration, are increasingly reported by the emerging field of cancer neuroscience (*Venkataramani et al., 2022*). Therefore, our results suggest that glucose mitochondrial metabolism mirrors GBM progression in mouse GL261 and CT2A models: more prevalent in smaller, well-perfused tumors, where glucose oxidation rate correlates with tumor cell proliferation (*Simões et al., 2022*); lower in larger, more poorly perfused

tumors, where recycling of the glutamate-glutamine pool may reflect a phenotype associated with secondary brain lesions.

Despite the excellent performance of tensor PCA denoising – threefold increase in SNR, approaching the original/raw values obtained previously with single-voxel $^2$H-MRS data (SNR ~20, *Simões et al., 2022*) – no further improvements in SNR could be achieved by free induction decay (FID) averaging within the tumor ROI (*Figure 1—figure supplement 8*). Therefore, further DGE-DMI preclinical studies aimed at detecting and quantifying relatively weak signals, such as tumor glutamate-glutamine, and/or increasing the nominal spatial resolution to better correlate those metabolic results with histology findings (e.g. in the tumor margin), should improve basal SNR with higher magnetic field strengths, more sensitive RF coils, and advanced DMI pulse sequences *Peters et al., 2021*. In the kinetic model, the extracellular volume fraction was fixed to ensure model stability, as previously demonstrated using the tumor average across all subjects (*Simões et al., 2022*). This approximation may not fully reflect the intra- and inter-tumor heterogeneity of this parameter in both cohorts, and may not be representative of its peritumoral regions. Still, we opted for this approach, rather than pixel-wise adjustments according to DGE-T1 extracellular volume fraction maps, given (i) the relative insensitivity of the model to the actual extracellular volume fraction value used (*Simões et al., 2022*), also verified in the present study (*Figure 1—figure supplement 9*); and particularly, because (ii) we did not have DCE-T1 data for the full cohort, thus it was not feasible to perform individual corrections, which in any case would ultimately be prone to error at tumor periphery/border regions, where exact delimitations are typically debatable. Finally, our results are indicative of higher microglia/macrophage infiltration in CT2A than in GL261 tumors, which is inconsistent with another study reporting higher immunogenicity of GL261 tumors than CT2A for microglia and macrophage populations (*Khalsa et al., 2020*). Such discrepancy could be related to methodologic differences between the two studies, namely the endpoint-guided assessment of tumor growth (bioluminescence vs MRI, more precise volumetric estimations) and tumor stage (GL261 at 23–28 vs 16–18 d post-injection, i.e. less time for an immune cell to infiltration in our case), presence/absence of a cell transformation step (GFP-Fluc engineered vs we used original cell lines), or perhaps media conditioning effects during cell culture due to the different formulations used (DMEM vs RPMI).

Our results clearly highlight the importance of mapping pathway fluxes alongside de novo concentrations to improve the characterization of the complex and dynamic heterogeneity of GBM metabolism. This may be a determinant for the longitudinal assessment of GBM progression, with end-point validation; and/or treatment-response, to help select among new therapeutic modalities targeting GBM metabolism (*Molina et al., 2018*; *Shi et al., 2019*) or monitoring the efficacy of novel immunotherapy approaches (*Wang et al., 2024*) beyond conventional chemoradiotherapy (*Low et al., 2024*). Importantly, DGE-MRI has already been demonstrated in glioma patients with i.v. administration of glucose using Chemical Exchange Saturation Transfer (glucoCEST) and relaxation-based methods (*Paech et al., 2017*; *Mo et al., 2025*), to map the spatiotemporal kinetics of glucose accumulation rather than quantifying its downstream metabolic fluxes through glycolysis and mitochondrial oxidation, as we did. The latter could potentially benefit from an improved kinetic model simultaneously assessing cerebral glucose and oxygen metabolism, as recently demonstrated in the rat brain with a combination of $^2$H and $^{17}$O MR spectroscopy (*Zhang et al., 2025*). Moreover, DMI has been demonstrated on a 9.4T clinical MRI scanner (*Ruhm et al., 2021*), benefiting from the higher sensitivity in the much larger human brain compared to mice: 200 cm$^3$ (*Yu et al., 2014*) and 415 mm$^3$ (*Kovacević et al., 2005*), respectively.

In summary, we report a DGE-DMI method for quantitative mapping of glycolysis and mitochondrial oxidation fluxes in mouse GBM, highlighting its importance for metabolic characterization and potential for in vivo GBM phenotyping in different models and progression stages. In large mouse GBM tumors, lactate metabolism underlies model-specific features, consistent with faster turnover in more disrupted stromal-vascular phenotypes and mirroring intra-tumor gradients of cell density and proliferation, whereas recycling of the glutamate-glutamine pool may reflect a phenotype associated with secondary brain lesions. Tensor PCA denoising significantly improved spectral signal-to-noise, which helped reveal such associations between regional glucose metabolism and phenotypic features of intra- and inter-tumor heterogeneity. DGE-DMI is potentially translatable to high-field clinical MRI scanners for precision neuro-oncology imaging.

# Materials and methods

## Animals and cell lines

This study was performed in strict accordance with European Directive 2010/63 and the Portuguese law (Decreto-Lei 113/2013), following the FELASA (Federation of European Laboratory Animal Science Associations) guidelines and recommendations concerning laboratory animal welfare, and aligned with the ARRIVE (Animal Research: Reporting of In Vivo Experiments) guidelines. All animal experiments were performed at the Champalimaud Foundation Vivarium under project #05318, pre-approved by the competent institutional and national authorities: ORBEA (Champalimaud Foundation Animal Welfare Body) and DGAV (Direcção Geral de Alimentação e Veterinária), respectively. All the surgeries were performed under isoflurane anesthesia, and every effort was made to minimize suffering. A total of n=10 C57BL/6 j male mice were used in this study, bred at the Champalimaud Foundation Vivarium, and housed with ad libitum access to food and water and 12 hr light cycles. GL261 mouse glioma cells were obtained from the Tumor Bank Repository at the National Cancer Institute (Frederick MD, USA): 'GLIOMA 261,' sample number 0507815. CT2A mouse glioma cells were kindly provided by Prof. Thomas Seyfried at Boston College (Boston MA, USA). Both cell lines were grown in RPMI-1640 culture medium supplemented with 2.0 g/l Sodium Bicarbonate, 0.285 g/l L-glutamine, 10% Fetal Bovine Serum (Gibco), and 1% Penicillin-Streptomycin solution. The cell lines tested negative for mycoplasma contamination using the IMPACT Mouse FELASA 1 test (Idexx-BioResearch, Ludwigsburg, Germany).

## Glioma models

Tumors were induced in previously described (*Simões et al., 2008a*). Briefly, intracranial stereotactic injection of $1\times10^5$ GL261 or CT2A cells was performed in the caudate nucleus (n=5 and n=5 mice, respectively); analgesia (Meloxicam 1.0 mg/Kg s.c.) was administered 30 min before the procedure. Mice were anesthetized with isoflurane (1.5–2.0% in air) and immobilized on a stereotactic holder (Kopf Instruments, Tujunga/CA, USA) where they were warmed on a heating pad at 37 °C, while body temperature was monitored with a rectal probe (WPI ATC-2000, Hitchin, UK). The head was shaved with a small trimmer, and cleaned with iodopovidone, and the skull was exposed through an anterior-posterior incision in the midline with a scalpel. A 1 mm hole was drilled in the skull using a micro-driller, 0.1 mm posterior to the bregma and 2.32 mm lateral to the midline. The tumor cells ($1\times10^5$ in 4 µL PBS) were inoculated 2.35 mm below the cortical surface using a 10 µL Hamilton syringe (Hamilton, Reno NV, USA) connected to an automatic push-pull microinjector (WPI *Smartouch*, Sarasota FL, USA), by advancing the 26 G needle 3.85 mm from the surface of the skull (~1 mm skull-to-brain surface distance), pulling it back 0.5 mm, and injecting at 2 µL/min rate. The syringe was gently removed 2 min after the injection had finished, the skin sutured with surgical thread (5/0 braided silk, Ethicon, San Lorenzo Puerto Rico), and wiped with iodopovidone. During recovery from anesthesia, animals were kept warm on a heating pad and given an opioid analgesic (Buprenorphine 0.05 mg/Kg s.c.) before returning to their cage. Meloxicam analgesia was repeatedly administered at 24- and 48 hr post-surgery.

## In vivo studies

### Longitudinal MRI

GBM-bearing mice were imaged every 5–7 d on a 1 Tesla Icon MRI scanner (Bruker BioSpin, Ettlingen, Germany; running *ParaVision 6.0.1* software), to measure tumor volumes. For this, each mouse was placed in the animal holder under anesthesia (1–2% isoflurane in 31% O$_2$), heated with a recirculating water blanket, and monitored for rectal temperature (36–37°C) and breathing (60–90 BPM). Tumor volume was measured with T2-weighted [1]H-MRI (*RARE* sequence, 8x acceleration factor, repetition time TR = 2500 ms, echo time TE = 84 ms, 8 averages, 1 mm slice thickness, and 160×160 µm2 in-plane resolution), acquired in two orientations (coronal and axial). Each session lasted up to 30 min/animal.

### End-point MRI and DMI

GBM-bearing mice with tumors ≥35 mm3 (longitudinal MRI assessment) were scanned on a 9.4T BioSpec MRI scanner (Bruker BioSpin, Ettlingen, Germany; running under *ParaVision 6.0.1*), using a

[2]H/[1]H transmit-receive surface coil set customized for the mouse brain (NeosBiotec, Pamplona, Spain), as described before (*Simões et al., 2022*). Before each experiment, GBM-bearing mice fasted 4–6 hr, were weighed, and cannulated in the tail vein with a catheter connected to a home-built three-way injection system filled with: 6,6'-[2]H$_2$-glucose (1.6 M in saline); Gd-DOTA (25 mM in saline); and with heparinized saline (10 U/mL). Mice were placed on the animal holder under anesthesia (as in 2.3.1). Coilset quality factors (Q) for [1]H and [2]H channels were estimated in the scanner for each sample based on the ratio of the resonance frequency (400.34 and 61.45 MHz, for protons and deuterium, respectively) to its bandwidth (full width at half-minimum of the wobbling curve during the initial tuning adjustments): 175±8 and 200±12, respectively. Mice were imaged first with T2-weighted [1]H-MRI (*RARE* sequence, 8x acceleration factor, 3000ms TR, 40ms TE; 2 averages, 1 mm slice thickness, 70 μm in-plane resolution) in two orientations (coronal and axial). Then, the magnetic field homogeneity was optimized over the tumor region based on the water peak with [1]H-MRS (*STEAM* localization: 6×6×3 mm volume of interest, i.e. 108 μL) using localized first and second order shimming with the *MapShim Bruker* macro, leading to full widths at half-maximum (FWHM) of 28±5 Hz.

DMI was performed using a *slice-FID chemical-shift imaging* pulse sequence, with 175 ms TR, 256 spectral points sampled over a 1749 Hz window, and Shinnar-Le Roux RF pulse (*Shinnar et al., 1989*; *Pauly et al., 1991*) (0.42 ms, 10 kHz) with 55° flip angle, to excite a brain slice including the tumor: 18×18 mm field-of-view, and 2.27 mm slice thickness. After RF pulse calibration (using the natural abundance semi-heavy water peak, DHO), DGE-DMI data were acquired for 2h23min (768 repetitions), with i.v. bolus of 6,6'-[2]H$_2$-glucose (2 mg/g, 4 μL/g injected over 30 s; Euroisotop, St Aubin Cedex, France). Data were sampled with an 8×8 matrix and fourfold Fourier interpolated (*Vikhoff-Baaz et al., 2001*), rendering a 560 μm in-plane resolution. A reference T2-weighted image was additionally acquired with matching field-of-view and slice thickness, and 70 μm in-plane resolution.

Finally, animals underwent DCE T1-weighted [1]H-MRI (*FLASH* sequence, 8° flip-angle, 16ms TR, 4 averages, 150 repetitions, 1 slice with 140 μm in-plane resolution and 2.27 mm thickness, FOV size and position matching the DGE-DMI experiment), with i.v. bolus injection of Gd-DOTA (0.1 mmol/Kg, injected over 30 s; Guerbet, Villepinte, France). Animals were then sacrificed, brains were removed, washed in PBS, and immersed in 4% PFA.

## MRI/DMI Processing

### T2-weighted [1]H-MRI

T2-weighted MRI data were processed in ImageJ 1.53 a (Rasband, W.S., ImageJ, U. S. National Institutes of Health, Bethesda, Maryland, USA, https://imagej.nih.gov/ij/, 1997–2018). For each animal, the tumor region was manually delineated on each slice, and the sum of the areas multiplied by the slice thickness to estimate the volume, which was averaged across the two orientations acquired (coronal and axial).

### DGE-DMI

DGE-DMI data were processed in MATLAB R2018b (Natick, Massachusetts: The MathWorks Inc) and jMRUI 6.0b (*Stefan et al., 2009*). Each dataset was averaged to 12 min temporal resolution and noise regions outside the brain, as well as the olfactory bulb and cerebellum, were discarded, rendering a 4D spectral-spatial-temporal matrix of 256×32×32×12 points. After automated phase-correction of each spectrum, the 4D matrix was denoised with a tensor PCA denoising approach (*Olesen et al., 2023*). For this, a (*Lu et al., 2010*; *Lu et al., 2010*; *Lu et al., 2010*) window and tensor structure [1 2:3 4] were used for patch processing the spectral, spatial, and temporal dimensions with, whereas the a priori average standard deviation of the noise in each spectrum (calculated σ$^2$) was used to avoid deleterious effects of spatially-correlated noise (*Henriques et al., 2023*). Then, these denoised spectra were analyzed voxel-wise by individual peak fitting with AMARES (similarly to the single-spectrum analysis reported previously in *Simões et al., 2015*), using a basis set for DHO (4.76 ppm: short- and long-T2 fractions *De Feyter et al., 2018*) and deuterium-labeled: glucose (Glc, 3.81 ppm), glutamate-glutamine (Glx, 2.36 ppm), and lactate (Lac, 1.31 ppm); relative linewidths referenced to the estimated short-T2 fraction of DHO, according to the respective T2 relaxation times reported by *De Feyter et al., 2018*. The natural abundance DHO peak (DHOi) was further used to select and quantify both original and denoised spectra: SNR$_{DHOi}$ >3.5 and 13.88 mM reference (assuming 80% water content in the brain and 0.03% natural abundance of DHO), respectively. Metabolite concentrations

(CRLB <50%; otherwise discarded) were corrected for T1 and labeling-loss effects, according to the values reported by de Feyter et al (T1, ms: DHO, 320; Glc, 64; Glx, 146; Lac, 297) (*De Feyter et al., 2018*) and de Graaf et al (number of magnetically equivalent deuterons: DHO, 1; Glc, 2; Glx, 1.2; Lac, 1.7) (*de Graaf et al., 2021*), respectively. Thus, the concentration of each metabolite (m) at each time point was estimated as (*Equation 1*):

$$Conc_m = \frac{Area_m - Area0_m}{d_m} \times \frac{C_{DHO}}{C_m} \times \frac{d_{DHO}}{Area0_{DHO}} \times Conc_{ref} \tag{1}$$

Area=peak area; *Area0*=average peak area before injection; *d*=number of magnetically equivalent deuterons corrected for labeling-loss effects; *C*=T1 correction factor (1-exp(-TR/T1)); and $Conc_{ref}$ = reference DHO concentration.

The time-course changes of $^2$H-labeled metabolite (Glc, Glx, and Lac) concentrations were fitted using a modified version of the kinetic model reported by *Kreis et al., 2020*, to estimate the maximum rate of Glc consumption (total, $V_{max}$) for Glx synthesis (mitochondrial oxidation, $V_{glx}$) and Lac synthesis (glycolysis, $V_{lac}$), and the confidence intervals for all estimated parameters:

$$V_{max} = V_{lac} + V_{glx} \tag{2}$$

The coupled differential equations describing the concentration kinetics of each metabolite were:

$$\frac{d[Glc]}{dt} = k_g \left( C_p - \frac{[Glc]}{v} \right) - f \left( \frac{V_{max}[Glc]}{f.v.k_m + [Glc]} \right) \tag{3}$$

$$\frac{d[Lac]}{dt} = \frac{fV_{lac}[Glc]}{f.v.k_m + [Glc]} - k_{lac}[Lac] \tag{4}$$

$$\frac{d[Glx]}{dt} = \frac{fV_{glx}[Glc]}{f.v.k_m + [Glc]} - k_{glx}[Glx] \tag{5}$$

where: $k_g$, apparent rate constant of glucose transfer between blood and tumor (min$^{-1}$); $k_{glx}$, apparent rate constant of Glx elimination (min$^{-1}$); $k_{lac}$, apparent rate constant of lactate elimination (min$^{-1}$); $C_p = a_1 \cdot e^{-k_p \cdot t}$, Glc concentration in plasma (mM); $a_1$, the Glc concentration after the bolus injection (mM); and $k_p$, the effective rate constant of labeled glucose transfer to tissue (min$^{-1}$). As reported previously (*Simões et al., 2022*), the following parameters were fixed: fraction of deuterium enrichment (*f*), at 0.6 (*Kreis et al., 2020*); constant for glucose uptake ($k_m$), at 10 mM (*Marín-Hernández et al., 2011*; *Williams et al., 2012*); and the extravascular-extracellular volume fraction (*v*), at 0.22 – average estimation from DCE-T1-weighted MRI analysis (*Supplementary file 1a, table 1*). All the other parameters were fitted without any restrictions to their range. Metabolic rate maps were displayed and analyzed pixel-wise using cut-off points defined by five times their respective confidence intervals.

## DCE T1-weighted MRI

DCE T1-weighted MRI data were processed with DCE@urLab (*Ortuño et al., 2013*), as before (*Simões et al., 2022*). First, ROIs were manually delineated for each tumor and the time-course data was fitted with the Extended Tofts 2-compartment model *Tofts, 1997*, to derive the volume transfer constant between plasma and tumor extravascular-extracellular space ($k^{trans}$), the washout rate between extravascular-extracellular space and plasma ($k_{ep}$), and the extravascular-extracellular volume fraction ($v_e$). Then, each dataset was reprocessed by down-sampling the original in-plane resolution to match the DGE-DMI experiment (0.56×0.56×2.27 mm$^3$), and fitting the time-course data pixel-wise with the Extended Tofts 2-compartment model to derive $k^{trans}$, $k_{ep}$, and $v_e$ maps (pixels with root-mean square error >0.005 discarded).

## Histopathology and immunohistochemistry

Whole brains fixed in 4% PFA were embedded in paraffin and sectioned at 30 different levels on the horizontal plane, spanning the whole tumor area. 4 µm sections were stained with H&E (Sigma-Aldrich, St. Louis MO, USA), digitized (Nanozoomer, Hamamatsu, Japan), and analyzed by an experimental pathologist blinded to experimental groups, according to previously established criteria (*Simões et al., 2022*). Then, QuPath v0.4.3 built-in tools (*Bankhead et al., 2017*) were used to highlight

different tumor regions: Tumor ROIs, corresponding to the bulk tumor, were delineated first with 'create threshold' and then manually corrected; P-Margin ROIs, including areas of peritumoral infiltration, were delineated with 'expand annotations' by expanding 100 µm the tumor margin toward the adjacent brain parenchyma; Infiltrative ROIs, corresponding to specific infiltrative regions, were manually annotated. Between 3–6 sections of each tumor were also immunostained for Ki67 (mouse anti-ki67, BD, San Jose CA, USA; blocking reagent, M.O.M ImmPRESS kit, Vector Laboratories, Burlingame CA, USA; liquid DAB$^+$, Dako North America Inc, Carpinteria CA, USA), digitized (Nanozoomer, Hamamatsu, Japan), and analyzed with QuPath built-in tools (*Bankhead et al., 2017*) for Tumor and P-Margin ROIs, defined as detailed above. Thus, Ki67$^{+/-}$ cells were counted semi-automatically to determine the total number of cells, the cell density, and the proliferation index (% Ki67$^+$ cells) as the average across slices for each ROI, and respective Tumor/P-Margin ratios. This procedure was repeated for each animal. In addition, one histologic section corresponding to each DGE-DMI slice was immunostained for Iba-1 (rabbit anti-Iba-1, Fujifilm Wako PCC, Osaka, Japan; NovolinkTM Polymer, Leica Biosystems, UK; liquid DAB+, Dako North America Inc, Carpinteria CA, USA), digitalized (Philips UFS v1.8.6614 slide scanner) and analyzed in QuPath. Tumor region and peritumoral margin regions were automatically annotated as outlined above, and Iba-1 positive staining was quantified across all annotations using the threshold tools, adjusted for each slide to account for variations in staining intensity, to calculate the percentage of Iba-1 positive area: (Iba-1 +area/total annotation area)×100.

## Statistical analyses

Data were analyzed in MATLAB R2018b (Natick, Massachusetts: The MathWorks Inc) using the two-tailed Student's *t*-test, either unpaired (comparing different animal cohorts) or paired (comparing the same animal cohort in different conditions). Differences at the 95% confidence level (p=0.05) were considered statistically significant. Correlation analyses were carried out with the Pearson R coefficient. Error bars indicate standard deviation unless indicated otherwise.

## Acknowledgements

This work was supported by: H2020-MSCA-IF-2018, 844776 (RVS); FCT CEEC-IND4ed, ref 2021.02777. CEECIND/CP1675/CT0003 (RNH); and the Champalimaud Foundation. The authors thank Dr. Thomas Seyfried for access to the CT2A cell line and helpful discussion about the manuscript, and the Vivarium of the Champalimaud Centre for the Unknown, a research infrastructure of CONGENTO co-financed by Lisbon Regional Operational Programme (Lisboa2020), under the PORTUGAL 2020 Partnership Agreement, through the European Regional Development Fund (ERDF) and Fundação para a Ciência e Tecnologia (Portugal), under the project LISBOA-01–0145-FEDER-022170.

## Additional information

### Funding

| Funder | Grant reference number | Author |
| --- | --- | --- |
| H2020 Marie Skłodowska-Curie Actions | 10.3030/844776 | Rui Vasco Simoes |
| Fundação para a Ciência e a Tecnologia | 10.54499/2021.02777.ceecind/cp1675/ct0003 | Rafael Neto Henriques |
| Fundação Champalimaud | internal | Noam Shemesh |

The funders had no role in study design, data collection and interpretation, or the decision to submit the work for publication.

### Author contributions

Rui Vasco Simoes, Conceptualization, Resources, Data curation, Formal analysis, Supervision, Funding acquisition, Validation, Investigation, Visualization, Methodology, Writing – original draft, Project administration, Writing – review and editing; Rafael Neto Henriques, Software, Investigation, Visualization, Methodology, Writing – review and editing; Jonas L Olesen, Sune N Jespersen, Software,

Visualization, Methodology, Writing – review and editing; Beatriz M Cardoso, Investigation, Visualization, Writing – review and editing; Francisca F Fernandes, Software, Formal analysis, Visualization, Writing – review and editing; Mariana AV Monteiro, Investigation, Visualization, Methodology, Writing – review and editing; Tânia Carvalho, Formal analysis, Investigation, Visualization, Methodology, Writing – review and editing; Noam Shemesh, Supervision, Funding acquisition, Visualization, Writing – review and editing

## Author ORCIDs
Rui Vasco Simoes https://orcid.org/0000-0001-7574-4723
Noam Shemesh https://orcid.org/0000-0001-6681-5876

## Ethics

This study was performed in strict accordance with European Directive 2010/63 and the Portuguese law (Decreto-Lei 113/2013), following the FELASA (Federation of European Laboratory Animal Science Associations) guidelines and recommendations concerning laboratory animal welfare, and aligned with the ARRIVE (Animal Research: Reporting of In Vivo Experiments) guidelines. All animal experiments were performed at the Champalimaud Foundation Vivarium under project #05318, pre-approved by the competent institutional and national authorities: ORBEA (Champalimaud Foundation Animal Welfare Body) and DGAV (Direção Geral de Alimentação e Veterinária), respectively. All the surgeries were performed under isoflurane anesthesia, and every effort was made to minimize suffering.

Reviewer #1 (Public review): https://doi.org/10.7554/eLife.100570.3.sa1
Reviewer #3 (Public review): https://doi.org/10.7554/eLife.100570.3.sa2
Author response https://doi.org/10.7554/eLife.100570.3.sa3

---

## Additional files

### Supplementary files
Supplementary file 1. Supplementary tables.

MDAR checklist

### Data availability
All data generated or analyzed during this study are included in the manuscript and supporting files; source data files are publicly available - deposited in Dryad under https://doi.org/10.5061/dryad.905qfttwb (Simoes_eLife2024_Data.zip).

The following dataset was generated:

| Author(s) | Year | Dataset title | Dataset URL | Database and Identifier |
|---|---|---|---|---|
| Simoes RV | 2025 | Data from: Deuterium metabolic imaging phenotypes mouse glioblastoma heterogeneity through glucose turnover kinetics | https://doi.org/10.5061/dryad.905qfttwb | Dryad Digital Repository, 10.5061/dryad.905qfttwb |

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
