## [Editor Report · eLife Assessment]

This study provides a **valuable** approach to image and analyze in vivo metabolic flux through glucose turnover kinetics in glioblastoma tumor microenvironments. The evidence for the method's validity is **convincing**, which establishes the dynamic Deuterium Metabolic Imaging technique as an effective tool enabling non-invasive exploration of various tumors.

---

## [Referee Report · Reviewer #1 (Public review)]

In the resubmission Simões et al. emphasize the efficacy of their novel, non-invasive imaging methodology in mapping glucose-kinetics to predict key tumor features in two commonly used syngeneic mouse models of glioblastoma. The authors highlight that DGE-DMI has the potential to capture metabolic fluxes with greater sensitivity and acknowledge that future validation of DGE-DMI in patient-derived and spontaneous GBM models, as well as in the context of genetic manipulation of metabolism, would strengthen its clinical application. To further demonstrate the ability of DGE-DMI to predict tumor features, they included an assessment of myeloid cell infiltration along with proliferation, peritumoral invasion, and distant migration. Overall, the authors offer a novel method to the scientific community that can be further tested and adapted for interrogating GBM heterogeneity.

---

## [Referee Report · Reviewer #3 (Public review)]

Summary:

Simoes et al enhanced dynamic glucose-enhanced (DGE) deuterium spectroscopy with Deuterium Metabolic Imaging (DMI) to characterize the kinetics of glucose conversion in two murine models of glioblastoma (GBM). The authors combined spectroscopic imaging and noise attenuation with histological analysis and showcased the efficacy of metabolic markers determined from DGE DMI to correlate with histological features of the tumors. This approach is also potent to differentiate the two models from GL261 and CT2A.

Strengths:

The primary strength of this study is to highlight the significance of DGE DMI to interrogate the metabolic flux from glucose. The authors focused on glutamine/glutamate and lactate. They attempted to correlate the imaging findings with in-depth histological analysis to depict the link between metabolic features and pathological characteristics such as cell density, infiltration, and distant migration.

---

## [Author Response]

The following is the authors’ response to the original reviews.

**eLife assessment**
This work describes a convincingly validated non-invasive tool for in vivo metabolic phenotyping of aggressive brain tumors in mice brains. The analysis provides a valuable technique that tackles the unmet need for patient stratification and hence for early assessment of therapeutic efficacy. However, wider clinical applicability of the findings can be attained by expanding the work to include more diverse tumor models.

We thank the Editors for their comments. This concern was also raised by Reviewer 1 in the Public Review, where we address in more detail – please refer to comment PR-R1.C1. In brief, we agree that a more clinically relevant model should provide more translatable results to patients, and acknowledge this better in the revised manuscript: page 18 (lines 14-17), “While patient-derived xenografts and de novo models would be more suited to recapitulate human GBM heterogeneity and infiltration features, and genetic manipulation of glycolysis and mitochondrial oxidation pathways potentially relevant to ascertain DGE-DMI sensitivity for their quantification, (…)”. However, we also believe that the potential of DGE-DMI for application to different glioblastoma models or patients is demonstrated clearly enough with the two immunocompetent models we chose, extensively reported in the literature as reliable models of glioblastoma.

**Public Reviews:**

**Reviewer #1 (Public Review):**
Summary:This work introduces a new imaging tool for profiling tumor microenvironments through glucose conversion kinetics. Using GL261 and CT2A intracranial mouse models, the authors demonstrated that tumor lactate turnover mimicked the glioblastoma phenotype, and differences in peritumoral glutamate-glutamine recycling correlated with tumor invasion capacity, aligning with histopathological characterization. This paper presents a novel method to image and quantify glucose metabolites, reducing background noise and improving the predictability of multiple tumor features. It is, therefore, a valuable tool for studying glioblastoma in mouse models and enhances the understanding of the metabolic heterogeneity of glioblastoma.Strengths:By combining novel spectroscopic imaging modalities and recent advances in noise attenuation, Simões et al. improve upon their previously published Dynamic Glucose-Enhanced deuterium metabolic imaging (DGE-DMI) method to resolve spatiotemporal glucose flux rates in two commonly used syngeneic GBM mouse models, CT2A and GL261. This method can be standardized and further enhanced by using tensor PCA for spectral denoising, which improves kinetic modeling performance. It enables the glioblastoma mouse model to be assessed and quantified with higher accuracy using imaging methods.The study also demonstrated the potential of DGE-DMI by providing spectroscopic imaging of glucose metabolic fluxes in both the tumor and tumor border regions. By comparing these results with histopathological characterization, the authors showed that DGE-DMI could be a powerful tool for analyzing multiple aspects of mouse glioblastoma, such as cell density and proliferation, peritumoral infiltration, and distant migration.Weaknesses:(1) Although the paper provides clear evidence that DGE-DMI is a potentially powerful tool for the mouse glioblastoma model, it fails to use this new method to discover novel features of tumors. The data presented mainly confirm tumor features that have been previously reported. While this demonstrates that DGE-DMI is a reliable imaging tool in such circumstances, it also diminishes the novelty of the study.

PR-R1.C1 – We thank the Reviewer for the detailed analysis and reply below to each point. PR-R1.C1.1 - novelty: We thank the Reviewer for the comments and understand their perspective. While we acknowledge that our paper is more methodologically oriented, we also believe that significant methodological advances are critical for new discoveries. This was our main motivation and is demonstrated in the present work, showing the ability to map in vivo metabolic fluxes in mouse glioma, a “hot topic” and very desirable in the cancer field.

PR-R1.C1.2 – additional tumor features: To strengthen the biological relevance of this methodologic novelty, we have now included immune cell infiltration among the tumor features assessed, besides perfusion, histopathology, cellularity and cell proliferation. For this, we performed iba-1 immunostaining for microglia/ macrophages, now included in Fig. 2-B. These new results demonstrate significantly higher microglia/macrophage infiltration in CT2A tumors compared to GL261, particularly at the tumor border. This is very consistent with the respective tumor phenotypes, namely differences in cell density and cellularity between the 2 cohorts and across pooled cohorts, as we now report: page 9 (lines 10-18), “Such phenotype differences were reflected in the regional infiltration of microglia/macrophages: significantly higher at the CT2A peritumoral rim (PT-Rim) compared to GL261, and slightly higher in the tumor region as well (Fig 2B). Further quantitative regional analysis of Tumor-to-PT-Rim ROI ratios revealed: (i) 47% lower cell density (p=0.004) and 32% higher cell proliferation (p=0.026) in GL261 compared to CT2A (Fig 2C, Table S3); and (ii) strong negative correlations in pooled cohorts between microglia/macrophage infiltration and cellularity (R=-0.91, p=<0.001) or cell density (R=-0.77, p=0.016), suggesting more circumscribed tumor growth with higher peripheral/peritumoral infiltration of immune cells.”; and page 16 (lines 13-19), “GL261 tumors were examined earlier after induction than CT2A (17±0 vs. 30±5 days, p = 0.032), displaying similar volumes (57±6 vs. 60±14, p = 0.813) but increased vascular permeability (8.5±1.1 vs 4.3±0.5 10^3^/min: +98%, p=0.001), more disrupted stromal-vascular phenotypes and infiltrative growth (5/5 vs 0/5), consistent with significantly lower tumor cell density (4.9±0.2 vs. 8.2±0.3 10^-3^ cells/µm^2^: -40%, p<0.001) and lower peritumoral rim infiltration of microglia/macrophages (2.1±0.7 vs. 10.0±2.3 %: -77%, p=0.008)”.

PR-R1.C1.3 – new tumor features and DGE-DMI: Importantly, such regional differences in cellularity/cell density and immune cell infiltration between the two cohorts were remarkably mirrored by the lactate turnover maps (Fig 3-C), as we now report in the manuscript: page 12 (lines 6-15), “GL261 tumors accumulated significantly less lactate in the core (1.60±0.25 vs 2.91±0.33 mM: -45%, p=0.013) and peritumor margin regions (0.94±0.09 vs 1.46±0.17 mM: 36%, p=0.025) than CT2A – Fig 3 A-B, Table S1. Consistently, tumor lactate accumulation correlated with tumor cellularity in pooled cohorts (R=0.74, p=0.014). Then, lower tumor lactate levels were associated with higher lactate elimination rate, k_lac_ (0.11±0.1 vs 0.06±0.01 mM/min: +94%, p=0.006) – Fig 3B – which in turn correlated inversely with peritumoral rim infiltration of microglia/macrophages in pooled cohorts (R=-0.73, p=0.027) – Fig 3-C. Further analysis of Tumor/P-Margin metabolic ratios (Table S3) revealed: (i) +38% glucose (p=0.002) and -17% lactate (p=0.038) concentrations, and +55% higher lactate consumption rate (p=0.040) in the GL261 cohort; and (ii) lactate ratios across those regions reflected the respective cell density ratios in pooled cohorts (R=0.77, p=0.010) – Fig 3-C”. This is a novel, relevant feature compared to our previous work, as highlighted in our discussion: page 17 (lines 1-8), “Tumor vs peritumor border analyses further suggest that lactate metabolism reflects regional histologic differences:

lactate accumulation mirrors cell density gradients between and across the two cohorts; whereas lactate consumption/elimination rate coarsely reflects cohort differences in cell proliferation, and inversely correlates with peritumoral infiltration by microglia/macrophages across both cohorts. This is consistent with GL261’s lower cell density and cohesiveness, more disrupted stromal-vascular phenotypes, and infiltrative growth pattern at the peritumor margin area, where less immune cell infiltration is detected and relatively lower cell division is expected [43]”.

We trust that these new features recovered from DGE-DMI (Fig 2-B and Fig 3-C) show its potential for new discoveries in glioblastoma.

(2) When using DGE-DMI to quantitatively map glycolysis and mitochondrial oxidation fluxes, there is no comparison with other methods to directly identify the changes. This makes it difficult to assess how sensitive DGE-DMI is in detecting differences in glycolysis and mitochondrial oxidation fluxes, which undermines the claim of its potential for in vivo GBM phenotyping.

PR-R1.C2: We thank the reviewer for raising this important point. The validity of the method for mapping specific metabolic kinetics in mouse glioma was reported in our previous work, using the same animal models, as specified in the introduction (page 4, lines 10-13): “we recently (…) propose[d] Dynamic Glucose-Enhanced (DGE) 2H-MRS [31], demonstrating its ability to quantify glucose fluxes through glycolysis and mitochondrial oxidation pathways in vivo in mouse GBM (…)”. Therefore, this was not reproduced in the present work.

In brief, our DGE-DMI results are very consistent with our previous study, where DGE single voxel deuterium spectroscopy was performed in the same tumor models with higher temporal resolution and SNR (as state on page 16, lines 9-10: glycolytic lactate synthesis rate, 0.59±0.04 vs. 0.55±0.07 mM/min; glucose-derived glutamate-glutamine synthesis rate, 0.28±0.06 vs. 0.40±0.08 mM/min), which in turn matched well the values reported by others for glucose consumption rate through:

(i) glycolysis, in different tumor models including mouse lymphoma in vivo (0.99 mM/min, by DGE-DMI Kreis et al. 2020), rat breast carcinoma in situ (1.43 mM/min, using a biochemical assay Kallinowski et al. 1988), and even perfused GBM cells (1.35 fmol min^−1^ cell^−1^, according to Hyperpolarized 13C-MRS Jeong et al. 2017), very similar to our previous in vivo measurements in GL261 tumors: 0.50 ± 0.07 mM min^−1^ = 1.25 ± 0.16 fmol min^−1^ cell^−1^ (Simoes et al. 2022);

(ii) mitochondrial oxidation, very similar to previous in vivo measurements in mouse GBM xenografts (0.33 mM min^−1^, using 13C spectroscopy Lai et al. 2018), and particularly to our in situ measurements in cell culture for (GL261, 0.69 ± 0.09 fmol min^−1^ cell^−1^; and CT2A 0.44 ± 0.08 fmol min^−1^ cell^−1^), remarkably similar to the in vivo measurements in the respective tumors in vivo (Gl261, 0.32 ± 0.10 mM min^−1^ = 0.77 ± 0.23 fmol min^−1^ cell^−1^; and CT2A, 0.51 ± 0.11 mM min^−1^ = 0.60 ± 0.12 fmol min^−1^ cell^−1^) (Simoes et al. 2022).

(3) The study only used intracranial injections of two mouse glioblastoma cell lines, which limits the application of DGE-DMI in detecting and characterizing de novo glioblastomas. A de novo mouse model can show tumor growth progression and is more heterogeneous than a cell line injection model. Demonstrating that DGE-DMI performs well in a more clinically relevant model would better support its claimed potential usage in patients.

PR-R1.C3: We agree that a more clinically relevant model, such as the one suggested by the Reviewer, would in principle be better suited to provide more translatable results to patients. We however believe that the potential of DGE-DMI for application to different glioblastoma models or patients, with GBM or any other types of brain tumors for that matter, is demonstrated clearly enough with the two syngeneic models we chose, given their robustness and general acceptance in the literature as reliable immunocompetent models of GBM, and for their different histologic and metabolic properties. This way we could fully focus on the novel metabolic imaging method, as compared to our previous single-voxel approach. While both tumor cohorts (GL261 and CT2A) were studied at more advanced stages of tumor progression, the metabolic differences depicted are consistent with the histopathologic features reported, as discussed in the manuscript; namely, the lower glucose oxidation rates. We have now modified the manuscript to highlight this point: page 18 (lines 12-14), “While patient-derived xenografts and de novo models would be more suited to recapitulate human GBM heterogeneity and infiltration features, and genetic manipulation of glycolysis and mitochondrial oxidation pathways could be relevant to ascertain DGE-DMI sensitivity for their quantification, (…)”.

**Reviewer #2 (Public Review):**
Summary:In this work, the authors attempt to noninvasively image metabolic aspects of the tumor microenvironment in vivo, in 2 mouse models of glioblastoma. The tumor lesion and its surrounding appearance are extensively characterized using histology to validate/support any observations made with the metabolic imaging approach. The metabolic imaging method builds on a previously used approach by the authors and others to measure the kinetics of deuterated glucose metabolism using dynamic 2H magnetic resonance spectroscopic imaging (MRSI), supported by de-noising methods.Strengths:Extensive histological evaluation and characterization.Measurement of the time course of isotope labeling to estimate absolute flux rates of glucose metabolism.Weaknesses:(1) The de-noising method appears essential to achieve the high spatial resolution of the in vivo imaging to be compatible with the dimensions of the tumor microenvironment, here defined as the immediately adjacent rim of the mouse brain tumors. There are a few challenges with this approach. Often denoising methods applied to MR spectroscopy data have merely a cosmetic effect but the actual quantification of the peaks in the spectra is not more accurate than when applied directly to original non-denoised data. It is not clear if this concern is applicable to the denoising technique applied here. However, even if this is not an issue, no denoising method can truly increase the original spatial resolution at which data were acquired. A quick calculation estimates that the spatial resolution of the 2H MRSI used here is 30-40 times too low to capture the much smaller tumor rim volume, and therefore there is concern that normal brain tissue and tumor tissue will be the dominant metabolic signal in so-called tumor rim voxels. This means that the conclusions on metabolic features of the (much larger) tumor are much more robust than the observations attributed to the (much smaller) tumor microenvironment/tumor rim.

PR-R2.C1: We thank the Reviewer for the constructive comments regarding resolution and tumor rim, and denoising. These issues were raised more extensively in the section Recommendations For The Authors, where they are addressed in detailed (RA-R2.C2). In summary, we agree with the Reviewer that no denoising method can increase the nominal resolution; not was that our purpose. Thus, we clarify the relevance of spectral matrix interpolation in MRSI, and how our display resolution should in principle provide a better approximation to the ground truth than the nominal resolution, relevant for ROI analysis in the tumor margin. While we further show relevant correlations between metabolic maps and histologic features in tumor core and margin, we agree with the reviewer that our observations in the tumor core are more robust than those in the margin, and acknowledge this in the Discussion: page 19, lines 6-10: “Therefore, further DGE-DMI preclinical studies aimed at detecting and quantifying relatively weak signals, such as tumor glutamate-glutamine, and/or increase the nominal spatial resolution to better correlate those metabolic results with histology findings (e.g. in the tumor margin), should improve basal SNR with higher magnetic field strengths, more sensitive RF coils, and advanced DMI pulse sequences [55].”

(2) To achieve their goal of high-level metabolic characterization the authors set out to measure the deuterium labeling kinetics following an intravenous bolus of deuterated glucose, instead of the easier measurement of steady-state after the labeling has leveled off. These dynamic data are then used as input for a mathematical model of glucose metabolism to derive fluxes in absolute units. While this is conceptually a well-accepted approach there are concerns about the validity of the included assumptions in the metabolic model, and some of the model's equations and/or defining of fluxes, that seem different than those used by others.

PR-R2.C2: These concerns about the metabolic model, were also raised in more detail in the section Recommendations For The Authors, where they are addressed more extensively – please refer to RA-R2.C3 (glucose infusion protocol) and RA-R2.C4 (equations). In brief, we explain that the total volume injected (100uL/25g animal) is standard for i.v. administration in mice, and clarify this better in the manuscript (page 24, line 23); as well as the differences between our kinetic model and the original one reported by Kreis et al. (Radiology 2020), who quantified glycolysis kinetics on a subcutaneous mouse model of lymphoma, exclusively glycolytic and thus estimating the maximum glucose flux rate was from the lactate synthesis rate (Vmax = Vlac). Instead, we extended this model to account for glucose flux rates for lactate synthesis (Vlac) and also for glutamate-glutamine synthesis (Vglx) in mouse glioblastoma, where Vmax = Vlac + Vglx, also acknowledging its simplistic approach in the Discussion (page 20, lines 22-24: “(…) metabolic fluxes [estimations] through glycolysis and mitochondrial oxidation (…) could potentially benefit from an improved kinetic model simultaneously assessing cerebral glucose and oxygen metabolism, as recently demonstrated in the rat brain with a combination of 2H and 17O MR spectroscopy [62] (…)”).

**Reviewer #3 (Public Review):**
Summary:Simoes et al enhanced dynamic glucose-enhanced (DGE) deuterium spectroscopy with Deuterium Metabolic Imaging (DMI) to characterize the kinetics of glucose conversion in two murine models of glioblastoma (GBM). The authors combined spectroscopic imaging and noise attenuation with histological analysis and showcased the efficacy of metabolic markers determined from DGE DMI to correlate with histological features of the tumors. This approach is also potent to differentiate the two models from GL261 and CT2A.Strengths:The primary strength of this study is to highlight the significance of DGE DMI in interrogating the metabolic flux from glucose. The authors focused on glutamine/glutamate and lactate. They attempted to correlate the imaging findings with in-depth histological analysis to depict the link between metabolic features and pathological characteristics such as cell density, infiltration, and distant migration.Weaknesses:(1) A lack of genetic interrogation is a major weakness of this study. It was unclear what underlying genetic/epigenetic aberrations in GL261 and CT2A account for the metabolic difference observed with DGE DMI. A correlative metabolic confirmation using mass spectrometry of the two tumor specimens would give insight into the observed imaging findings.

PR-R3.C1: We thank the Reviewer for the helpful comments, which we break down below.

PR-R3.C1.1 - genetic interrogation/manipulation: While we did not have access to conditional models for key enzymes of each metabolic pathway, for their genetic manipulation, we did however assess the mitochondrial function in each cell line, showing a significantly higher respiration buffer capacity and more efficient metabolic plasticity between glycolysis and mitochondrial oxidation in GL261 cells compared to CT2A (Simoes et al. NIMG:Clin 2022). This could drive e.g. more active recycling of lactate through mitochondrial metabolism in GL261 cells, aligned with our observations of increased glucose-derived lactate consumption rate in those tumors compared to CT2A. We have now included this in the discussion (page 17, lines 812): “our results suggest increased lactate consumption rate (active recycling) in GL261 tumors with higher vascular permeability, e.g. as a metabolic substrate for oxidative metabolism [44] promoting GBM cell survival and invasion [45], aligned with the higher respiration buffer capacity and more efficient metabolic plasticity of GL261 cells than CT2A [31].”

PR-R3.C1.2 - correlation with post-mortem metabolic assessment: implementing this validation step would require an additional equipment, also not accessible to us: focalized irradiator, to instantly halt all metabolic reactions during animal sacrifice. We do believe that DGE-DMI could guide further studies of such nature, aimed at validating the spatio-temporal dynamics of regional metabolite concentrations in mouse brain tumors. Thus, the importance of end-point validation is now stressed more clearly in the manuscript (page 20, lines 13-16): “(…) mapping pathway fluxes alongside de novo concentrations (…) may be determinant for the longitudinal assessment of GBM progression, with end-point validation (…)”.

These concerns and recommendations were also raised by the Reviewer in the Recommendations to Authors section, where we address them more extensively – please see RA-R1.C3 and RA-R1.C2, respectively.

(2) A better depiction of the imaging features and tumor heterogeneity would support the authors' multimodal attempt.

PR-R3.C2: We agree with the Reviewer that including more imaging features would improve the non-invasive characterization of each tumor. Due to the RF coil design and time constraints, we did not acquire additional data, such as diffusion MRI to assess tissue microstructure. Instead, our multi-modal protocol included two dynamic MRI studies on each animal, for multiparametric assessment of tumor volume, metabolism and vascular permeability, using 1H-MRI, 2H-spectroscopy during 2H-labelled glucose injection, and 1H-imaging during Gd-DOTA injection, respectively. Rather than aiming at tumor radiomics, we focused on the dynamic assessment of tumor metabolic turnover with heteronuclear spectroscopy, which is challenging per se and particularly in mouse brain tumors, given their very small size. For such multi-modal studies we used a previously developed dual tuned RF coil: the deuterium coil (2H) positioned in the mouse head, for optimal SNR; whereas the proton coil (1H) had suboptimal performance compared a conventional single tuned coil, and was used only for basic localization and adjustments, reference imaging and tumor volumetry (T2-weighted), and DCE-T1 MRI (T1weighted). The latter was analyzed pixel-wise to assess spatial correlations between tumor permeability and metabolic metrics, as shown in Fig S3. Whereas the limited T2w MRI data collected was only analyzed for tumor volume assessment; no additional imaging features were extracted (e.g. kurtosis/skewness), since such assessment did not shown any differences between the two tumor cohorts in our previous study (Simoes et al NIMG:Clin 2022).

(3) Integration of the various cell types in the tumor microenvironment, as allowed with the resolution of DGE DMI, will explain the observed difference between GL261 and CT2A. Is there a higher percentage of infiltrative "other cells" observed in GL261 tumor?

PR-R3.C3: While DGE-DMI resolution is far larger than brain and brain tumor cell sizes, we now performed additional analysis to assess the percentage of microglia/macrophages in both cohorts. The results are now included in the manuscript, namely Fig. 2B, as previously explained in PR-R1.1. Interestingly though, we observed a lower percentage of infiltrative "other cells" in GL261 tumors compared to CT2A, which we discuss in the manuscript: pages 19-20 (lines 20-24 and 1-4), “Finally, our results are indicative of higher microglia/macrophage infiltration in CT2A than GL261 tumors, which is inconsistent with another study reporting higher immunogenicity of GL261 tumors than CT2A for microglia and macrophage populations [56]. Such discrepancy could be related to methodologic differences between the two studies, namely the endpointguided assessment of tumor growth (bioluminescence vs MRI, more precise volumetric estimations) and the stage when tumors were studied (GL261 at 23-28 vs 16-18 days postinjection, i.e. less time for immune cell to infiltration in our case), presence/absence of a cell transformation step (GFP-Fluc engineered vs we used original cell lines), or perhaps media conditioning effects during cell culture due to the different formulations used (DMEM vs RPMI).”

(4) This underlying technology with DGE DMI is capable of identifying more heterogeneous GBM tumors. A validation cohort of additional in vivo models will offer additional support to the potential clinical impact of this study.

PR-R3.C4: We agree with the Reviewer that applying DGE-DMI to more clinically-relevant models of human brain tumors will enhance its translational impact to patients, as also suggested by Reviewer 1 and addressed in PR-R1.C3. We also believe that the feasibility and potential of DGE-DMI for application to different glioblastoma models or patients, with GBM or any other primary or secondary brain tumors, is clearly demonstrated in our work, using two reliable and well-described immunocompetent models of GBM. In any case, we have now modified the manuscript to better acknowledge this point: page 18 (lines 14-16), “(…) patient-derived xenografts and de novo models would be more suited to recapitulate human GBM heterogeneity and infiltration features (…)”.

**Recommendations for the authors:**

**Reviewer #1 (Recommendations For The Authors):**
(1) The authors utilize longitudinal MRI to track tumor volumes but perform DMI at endpoint with late-stage tumors. Their previous publication applied metabolic imaging in tumors before the presence of necrosis. It would be valuable to perform longitudinal DMI to examine the evolution of glucose flux metabolic profile over time in the same tumor.

RA-R1.C1: We thank the Reviewer for the very useful comments to our manuscript. We agree – in this work, we aimed at “extending” our previous DGE-2H single-voxel methodology to multivoxel (DMI), thoroughly demonstrating (1) its in vivo application to the same immunocompetent models of glioblastoma and (2) the ability to depict their phenotypic differences, and therefore (3) the potential for the metabolic characterization of more advanced models of GBM and/or their progression stages. We believe these objectives were achieved. Our results indeed open several possibilities, from longitudinal assessment of the spatio-temporal metabolic changes during GBM progression (and treatment-response) to its application to other models recapitulating more closely the human disease. Now that we have comprehensively demonstrated a protocol for DGE-DMI acquisition, processing and analysis in mouse GBM (a very challenging methodology), and demonstrate it in different mouse GBM cell lines, new studies can be designed to tackle more specific questions, like the one suggested here by the Reviewer. We have modified the manuscript to make this point clearer: page 20 (lines 15-17), “This may be determinant for the longitudinal assessment of GBM progression, with end-point validation; and/or treatment-response, to help selecting among new therapeutic modalities targeting GBM metabolism (…)”; page 21 (lines 5-8), “(…) we report a DGE-DMI method for quantitative mapping of glycolysis and mitochondrial oxidation fluxes in mouse GBM, highlighting its importance for metabolic characterization and potential for in vivo GBM phenotyping in different models and progression stages.”.

(2) The authors demonstrate a promising correlation between metabolic phenotypes in vivo and key histopathological features of GBM at the endpoint. Directly assessing metabolites involved in glucose fluxes on endpoint tumor samples would strengthen this correlation.

RA-R1.C2: While we acknowledge the Reviewer’s point, there were two main limitations to implementing such validation step in our protocol:

(1) Since we performed dynamic experiments, at the end of each study most 2H-glucose-derived metabolites were already below their maximum concentration (or barely detectable in some cases), as depicted by the respective kinetic curves (Fig 1-D and Fig S7), and thus no longer detectable in the tissues. Importantly, DGE-DMI could guide further studies towards selecting the ideally time-point for validating different metabolite concentrations in specific brain regions.

(2) Such validation would require sacrificing the animals with a focalized irradiator (which we did not have), to instantly halt all metabolic reactions. Only then we could collect and analyze the metabolic profile of specific brain regions, either by in vitro MS or high-resolution NMR following extraction, or by ex vivo HRMAS analysis of the intact tissue, as reported previously by some of the authors for validation of glucose accumulation in different regions of mouse GL261 tumors (Simões et al. NMRB 2010: https://doi.org/10.1002/nbm.1421). Importantly, even if we did have access to a focalized irradiator, such protocols for metabolic characterization would compromise tissue integrity and thus the histopathologic analysis performed in this study.

We do agree with the importance of end-point validation and therefore stress it more clearly in the revised manuscript (page 20, lines 14-16): “(…) mapping pathway fluxes alongside de novo concentrations (…) may be determinant for the longitudinal assessment of GBM progression, with end-point validation (…)”.

(3) Genetic manipulation of key players in the metabolic pathways studied in this paper (glycolysis and mitochondrial oxidation) would offer a strong validation for the sensitivity of DGE-DMI in accurately distinguishing metabolites (lactate, glutamate-glutamine) and their dynamics.

RA-R1.C3: Thank you for this comment, we agree. This would be particularly relevant in the context of treatment-response monitoring. While such models were not available to us (conditional spatio-temporal manipulation of metabolic pathway fluxes), we believe our results can still demonstrate this point: We previously used in vivo DGE 2H-MRS to show evidence of decreased glucose oxidation fraction (Vglx/Vlac) in GL261 tumors under acute hypoxia (FiO2=12 %) compared to regular anesthesia conditions (FiO2=31 %), consistent with the inhibition of OXPHOS due to lower oxygens tensions (Simoes et al. NIMG:Clin 2022). In the present work, enhanced glycolysis in tumors vs peritumoral brain regions was clearly observed in all the animals studied, from both cohorts, as shown in Fig 1-B and Fig S4. Moreover, the spectral background (before glucose injection) is limited to a single peak in all the voxels: basal DHO, used as internal reference for spatio-temporal quantification of glucose, glutamine-glutamate, and lactate, all de novo and extensively characterized in healthy and glioma-bearing rodent brain (Lu et al. JCBFM 2018; Zhang et al. NMR Biomed 2024, de Feyter et al. SciAdv 2018; Batsios et al ClinCancerRes 2022; Simoes et al. NIMG:Clin 2022) and other rodent tumors (Kreis et al. Radiology 2020, Montrazi et al. SciRep 2023). We have modified the manuscript to clarify this point (page 18, lines 14-17) “(…) patient-derived xenografts and de novo models would be more suited to recapitulate human GBM heterogeneity and infiltration features, and genetic manipulation of glycolysis and mitochondrial oxidation pathways could be relevant to ascertain DGE-DMI sensitivity for their quantification (…)”.

(4) Please explain more why DEG-DMI can distinguish different glucose metabolites and how accurate it is.

RA-R1.C4: DGE-DMI is the imaging extension of our previous work based on single-voxel deuterium spectroscopy, therefore relying on the same fundamental technique and analysis pipeline but moving from a temporal analysis to a spatio-temporal analysis for each metabolite, and thus dealing with more data. Unlike conventional proton spectroscopy (1H), only metabolites carrying the deuterium label (2H) will be detected in this case, including the natural abundance DHO (~0.03%), the deuterated glucose injected and its metabolic derivatives, namely deuterated lactate and deuterated glutamate-glutamine. Due to their different molecular structures, the deuterium atoms will resonate at specific frequencies (chemical shifts, ppm) during a 2H magnetic resonance spectroscopy experiment, as illustrated in Fig 1-A. The method is fully reproducible and accurate, and has been extensively reported in the literature from high-resolution NMR spectroscopy to in vivo spectroscopic imaging of different nuclei, such as proton (1H), deuterium (2H), carbon (13C), phosphorous (31P), and fluorine (19F). Since the fundamental principles of DMI and its application to brain tumors have been very well described in the flagship article by de Feyter et al., we have now highlighted this in the manuscript: page 4 (lines 4-7), “Deuterium metabolic imaging (DMI) has been (…) demonstrated in GBM patients, with an extensive rationale of the technique and its clinical translation [18], and more recently in mouse models of patient-derived GBM subtypes (…)”.

(5) When mapping glycolysis and mitochondrial oxidation fluxes, add a control method to compare the reliability of DEG-DMI.

RA-R1.C5: This concern (“lack of a control method”) was also raised by the Reviewer in the section Public Reviews section, where we already address it (PR-R1.2).

(6) If using peritumoral glutamate-glutamine recycling as a marker of invasion capacity, what would be the correct rate of the presence of secondary brain lesions?

RA-R1.C6: While our results suggest the potential of peritumoral glutamate-glutamine recycling as a marker for the presence of secondary brain lesions, this remains to be ascertained with higher sensitivity for glutamate-glutamine detection. Therefore, we cannot make further conclusions in this regard.

To make this point clear, we state in different sections of the discussion: page 19 (lines 1-2), “(…) recycling of the glutamate-glutamine pool may reflect a phenotype associated with secondary brain lesions.”; and page 19 (lines 6-10), “Therefore, further DGE-DMI preclinical studies aimed at detecting and quantifying relatively weak signals, such as tumor glutamateglutamine, and/or increase spatial resolution to correlate those metabolic results with histology findings (e.g in the tumor margin), should improve basal SNR with higher magnetic field strengths, more sensitive RF coils, and advanced DMI pulse sequences [55]”.

(7) There are duplicated Vlac in Figure S3 B.

RA-R1.C7: This was a typo that has now been corrected. Thank you.

(8) Figure 4, it would be better to add a metabolic map of a tumor without secondary brain lesions to compare.

RA-R1.C8: We fully agree and have modified Fig 4 accordingly, together with its legend.

Particularly, we have included tumors C4 (without secondary lesions) vs G4 (with) for this “comparison”, since details of their histology, including the secondary lesions, are provided in Fig 2.

(9) Full name of SNR and FID should be listed when first mentioned.

RA-R1.C9: Agreed and modified accordingly, on pages 6-7 (lines 22-1), ”signal-to-noise-ratio (SNR)”, and page 19 (lines 5-6), “free induction decay (FID)”.

(10) Page 2, Line 14: (59{plus minus}7 mm3) is not needed in the abstract.

RA-R1.C10: As requested we have removed this specification from the Abstract.

(11) Page 4, Line 22: Closing out the Introduction section with a statement on broader implications of the present work would enhance the effectiveness of the section.

RA-R1.C11: We have added an additional sentence in this regard – pages 4-5 (lines 24-2): “Since DMI is already performed in humans, including glioblastoma patients [18], DGE-DMI could be relevant to improve the metabolic mapping of the disease.”

(12) Define all acronyms to facilitate comprehension. For example, principal component analysis (PCR) and signal-to-noise ratio (SNR).

R1.C12: Thank you for the comment. We have now defined all the acronyms when first used, including PCA (page 4 (line 11), “Marcheku-Pastur Principal Component Analysis (MP-PCA)”) and SNR (pages 6-7 (lines 22-1), as indicated above in comment R1.9).

(13) Some elements within the figures have lower resolution, specifically bar graphs.

RA-R1.C13: We apologize for this oversight. All the Figures have been revised accordingly, to correct this problem. Thank you.

(14) Page 13, Line 8: "underly" should be spelled "underlie."

RA-R1.C14: The typo has been corrected on page 15 (line 8), thank you.

(15) Page 14, Line 13: "better vascular permeability" would be more effectively phrased as "increased vascular permeability."

RA-R1.C15: This has also been corrected on page 16 (line 14), thank you.

**Reviewer #2 (Recommendations For The Authors):**
(1) I strongly suggest adding a scale bar in the histology figures.

RA-R2.C1: Thank you for spotting our oversight! This has now been added as requested to Fig 2.

(2) The 2H MRSI data were acquired at a nominal resolution of 2.25 x 2.27 x 2.25 mm^3, resulting in a nominal voxel volume of 11.5 uL. (In reality, this is larger due to the point spread function leading to signal bleeding from adjacent voxels). If we estimate the volume of the tumor rim, as indicated by the histology slides, as (generously) ~ 50 um in width, 3.2 mm long the diagonal of a 2.25 x 2.25 mm^2 square, and 2.27 mm high, we get a volume of 0.36 uL. Therefore the native spatial resolution of the 2H MRSI is at least 30 times larger than the volume occupied by the tumor rim/microenvironment. Normal tissue and tumor tissue will contribute the majority of the metabolic signal of that voxel. I feel an opposite approach could have been pursued: find out the spatial resolution needed to characterize the tumor rim based on the histology, then use a de-noising method to bring the SNR of those data to be acceptable. (this is just a thought experiment that assumes de-noising actually works to improve quantification for MRS data instead of merely cosmetically improve the data, so far the jury is still out on that, in my view).

RA-R2.C2 – We thank the Reviewer for the detailed analysis and reply below to each point.

RA-R2.C2.1 – spatial resolution and tumor rim: Our nominal voxel volume was indeed 11.5 uL, defined in-plane by the PSF which explains signal bleeding effects, as in any other imaging modality. The DMI raw data were Fourier interpolated before reconstruction, rendering a final in-plane resolution of 0.56 mm (0.72 uL voxel volume). The tumor rim (margin) analyzed was roughly 0.1 mm width (please note, not 0.05 mm), as explained in the methods section (page 28, line 16) and now more clearly defined with the scale bars in Fig 2. According to the Reviewer’s analysis, this would correspond to 0.1*3.2*2.27 = 0.73 uL, which we approximated with 1 voxel (0.72 uL), as displayed in Fig 3-A. Importantly, it has long been demonstrated that Fourier interpolation provides a better approximation to the ground truth compared to the nominal resolution, and even to more standard image interpolation performed after FT - see for instance Vikhoff-Baaz B et al. (MRI 2001. 19: 1227-1234), now citied in the Methods section: page 24, line 24 ([69]). While we do agree that both normal brain and tumor should contribute significantly to the metabolic signal in this relatively small region, we rely on extensive literature to maintain that despite its smoothing effect, the display resolution provides a better approximation to the ground truth and is therefore more suited than the nominal resolution for ROI analysis in this region. Still, we acknowledge this potential limitation in the Discussion: page 19, lines 6-10: “Therefore, further DGE-DMI preclinical studies aimed at detecting and quantifying relatively weak signals, such as tumor glutamate-glutamine, and/or increase the nominal spatial resolution to better correlate those metabolic results with histology findings (e.g. in the tumor margin), should improve basal SNR with higher magnetic field strengths, more sensitive RF coils, and advanced DMI pulse sequences [55].”

RA-R2.C2.2 – metabolic and histologic features at the tumor rim: Furthermore, we also performed ROI analysis of lactate metabolic maps in tumor and peritumoral rim areas closely reflected regional differences in cellularity and cell density, and immune cell infiltration between the 2 tumor cohorts and across pooled cohorts, as explained in the Public Review section - PR-R1.1 – and now report in the manuscript: page 12 (lines 6-16), “GL261 tumors accumulated significantly less lactate in the core (1.60±0.25 vs 2.91±0.33 mM: -45%, p=0.013) and peritumor margin regions (0.94±0.09 vs 1.46±0.17 mM: -36%, p=0.025) than CT2A – Fig 3 A-B, Table S1. Consistently, tumor lactate accumulation correlated with tumor cellularity in pooled cohorts (R=0.74, p=0.014). Then, lower tumor lactate levels were associated with higher lactate elimination rate, k_lac_ (0.11±0.1 vs 0.06±0.01 mM/min: +94%, p=0.006) – Fig 3B – which in turn correlated inversely with peritumoral margin infiltration of microglia/macrophages in pooled cohorts (R=-0.73, p=0.027) - Fig 3-C. Further analysis of Tumor/P-Margin metabolic ratios (Table S3) revealed: (i) +38% glucose (p=0.002) and -17% lactate (p=0.038) concentrations, and +55% higher lactate consumption rate (p=0.040) in the GL261 cohort; and (ii) lactate ratios across those regions reflected the respective cell density ratios in pooled cohorts (R=0.77, p=0.010) – Fig 3-C”; page 17 (lines 1-8), “Tumor vs peritumor border analyses further suggest that lactate metabolism reflects regional histologic differences: lactate accumulation mirrors cell density gradients between and across the two cohorts; whereas lactate consumption/elimination rate coarsely reflects cohort differences in cell proliferation, and inversely correlates with peritumoral infiltration by microglia/macrophages across both cohorts. This is consistent with GL261’s lower cell density and cohesiveness, more disrupted stromal-vascular phenotypes, and infiltrative growth pattern at the peritumor margin area, where less immune cell infiltration is detected and relatively lower cell division is expected [43]”.

RA-R2.C2.3 – alternative method: Regarding the alternative method suggested by the Reviewer, we have tested a similar approach in another region (tumor) and it did not work, as explained the Discussion section (page 19, lines 5-6) and Fig S11. Essentially, Tensor PCA performance improves with the number of voxels and therefore limiting it to a subregion hinders the results. In any case, if we understand correctly, the Reviewer suggests a method to further interpolate our data in the spatial dimension, which would deviate even more from the original nominal resolution and thus sounds counter-intuitive based on the Reviewer’s initial comment about the latter. More importantly, we would like to remark the importance of spectral denoising in this work, questioned by the Reviewer. There are several methods reported in the literature, most of them demonstrated only for MRI. We previously demonstrated how MPPCA denoising objectively improved the quantification of DCE-2H MRS in mouse glioma by significantly reducing the CRLBs: 19% improved fitting precision. In the present study, Tensor PCA denoising was applied to DGE-DMI, which led to an objective 63% increase in pixel detection based on the quality criteria defined, unambiguously reflecting the improved quantification performance due to higher spectral quality.

(3) Concerns re. the metabolic model: 2g/kg of glucose infused over 120 minutes already leads to hyperglycemia in plasma. Here this same amount is infused over 30 seconds... such a supraphysiological dose could lead to changes in metabolite pool sizes -which are assumed to not change since they are not measured, and also fractional enrichment which is not measured at all. Such assumptions seem incompatible with the used infusion protocol.

RA-R2.C3: We understand the concern. However, the protocol was reproduced exactly as originally reported by Kreis et al (Radiology 2020) that performed the measurements in mice and measured the fraction of deuterium enrichment (f=0.6). Since we also worked with mice, we adopted the same value for our model. The total volume injected was 100uL/25g animal, and adjusted for animal weight (96uL/24g average – Table S1), as we reported before (Simões et al. NIMG:Clin 2022), which is standard for i.v. bolus administration in mice as it corresponds to ~10% of the total blood volume. This volume is therefore easily diluted and not expected to introduce significant changes in the metabolic pool sizes. Continuous infusion protocols on the other hand will administer higher volumes, easily approaching the mL range when performed over periods as large as 120 min. This would indeed be incompatible with our bolus infusion protocol. We have now clarified this in the manuscript – page 24 (line 23): “i.v. bolus of 6,6^′2^H_2_-glucose 2 mg/g, 4 µL/g injected over 30 s (…)”.

(4) Vmax = Vlac + Vglx. This is incorrect: Vmax = Vlac.

RA-R2.C4: Thank you for raising this concern. As indicated in RA-R2.C3, our model (Simões et al. NIMG:Clin 2022) was adapted from the original model proposed by Kreis et al. (Radiology 2020), where the authors quantified glycolysis kinetics on a subcutaneous mouse model of lymphoma, exclusively glycolytic and thus estimating the maximum glucose flux rate was from the lactate synthesis rate (Vmax = Vlac). However, we extended this model to account for glucose flux rates for lactate synthesis (Vlac) and also for glutamate-glutamine synthesis (Vglx), where Vmax = Vlac + Vglx, as explained in our 2022 paper. While we acknowledge the rather simplistic approach of our kinetic model compared to others - reported by 13C-MRS under continuous glucose infusion in healthy mouse brain (Lai et al. JCBFM 2018) and mouse glioma (Lai et al. IJC 2018) – and acknowledge this in the Discussion (page 20, lines 22-24: “(…) metabolic fluxes [estimations] through glycolysis and mitochondrial oxidation (…) could potentially benefit from an improved kinetic model simultaneously assessing cerebral glucose and oxygen metabolism, as recently demonstrated in the rat brain with a combination of 2H and 17O MR spectroscopy [62] (…)”), our Vlac and Vglx results are consistent with our previous DGE 2H-MRS findings in the same glioma models, and very aligned with the literature, as discussed in PR-R1.C2.1.

(5) Some other items that need attention: 0.03 % is used as the value for the natural abundance of DHO. The natural abundance of 2H in water can vary somewhat regionally, but I have never seen this value reported. The highest seen is 0.015%.

RA-R2.C5: The Reviewers is referring to the natural abundance of deuterium in hydrogen: 1 in ~6400 is D, i.e. 0.015 %. The 2 hydrogen atoms in a water molecule makes ~3200 DHO, i.e. 0.03%. Indeed the latter can have slight variations depending on the geographical region, as nicely reported by Ge et al (Front Oncol 2022), who showed a 16.35 mM natural-abundance of DHO in the local tap water of St Luis MO, USA (55500/16.35 = 1/3364 = 0.034%).

(6) Based on the color scale bar in Figure 1, the HDO concentration appears to go as high as 30 mM. Even if this number is off because of the previous concern (HDO), it appears to be a doubling of the HDO concentration. Is this real? What would be the origin of that? No study using [6,6'-2H2]-glucose that I'm aware of has reported such an increase in HDO.

RA-R2.C6: As explained before (RA-R2.C3 and RA-R2.C4), we based our protocol and model on Kreis et al (Radiology 2020), who reported ~10 mM basal DHO levels raising up to ~27 mM after 90min, which are well within the ~30 mM ranges we report over a longer period (132 min).

Similar DHO levels were mapped with DGE-DMI in mouse pancreatic tumors (Montrazi et al. SciRep 2023).

(7) "...the central spectral matrix region selected (to discard noise regions outside the brain, as well as the olfactory bulb and cerebellum)". This reads as if k-space points correspond one-toone with imaging pixels, which is not the case.

RA-R2.C7: We rephrased the sentence to avoid such potential misinterpretation, specifically: page 25 (lines 19-21): “Each dataset was averaged to 12 min temporal resolution and the noise regions outside the brain, as well as the olfactory bulb and cerebellum, were discarded (…)”.

(8) The use of the term "glutamate-glutamine recycling" is not really appropriate since these metabolites are not individually detected with 2H MRS, which is a requirement to measure this neurotransmitter cycling.

RA-R2.C8: Thank you for this comment. To avoid this misinterpretation, we have now rephrased "glutamate-glutamine recycling" to “recycling of the glutamate-glutamine pool” in all the sentences, namely: page 2 (lines 14-15); page 15 (line 8); page 15 (line 8); page 19 (line 1); page 21 (line 10).

**Reviewer #3 (Recommendations For The Authors):**
(1) One major issue is the lack of underlying genetics, and therefore it is hard for readers to put the observed difference between GL261 and CT2A into context. The authors might consider perturbing the genetic and regulatory pathways on glycolysis and glutamine metabolism, repeating DGE DMI measure, in order to enhance the robustness of their findings.

RA-R3.C1: We thank the reviewer for the helpful revision and comments. The point made here is aligned with Reviewer 1’s, addressed in RA-R1.C3; and also with our previous reply to the Reviewer, PR-R3.C1. Thus, we agree that conditional spatio-temporal manipulation of metabolic pathway fluxes would be relevant to further demonstrate the robustness of DGEDMI, particularly for treatment-response monitoring. While such models were not available to us, our previous findings seem compelling enough to demonstrate this point. Thus, we previously showed a significantly higher respiration buffer capacity and more efficient metabolic plasticity between glycolysis and mitochondrial oxidation in GL261 cells compared to CT2A (Simoes et al. NIMG:Clin 2022), which could enhance lactate recycling through mitochondrial metabolism in GL261 cells and thus explain our observations of increased glucose-derived lactate consumption rate in those tumors compared to CT2A. We have now included this in the discussion (page 17, lines 8-12): “our results suggest increased lactate consumption rate (active recycling) in GL261 tumors with higher vascular permeability, e.g. as a metabolic substrate for oxidative metabolism [44] promoting GBM cell survival and invasion [45], aligned with the higher respiration buffer capacity and more efficient metabolic plasticity of GL261 cells than CT2A [31].” Moreover, we previously showed evidence of DGE-2H MRS’ ability to detect decreased glucose oxidation fraction (Vglx/Vlac) in GL261 tumors under acute hypoxia (FiO2=12 %) compared to regular anesthesia conditions (FiO2=31 %), consistent with the inhibition of OXPHOS due to lower oxygens tensions (Simoes et al. NIMG:Clin 2022).

(2) Is increased resolution possible for DGE DMI to correlate with histological findings?

RA-R3.C2: The resolution achieved with DGE DMI, or any other MRI method, is limited by the signal-to-noise ratio (SNR), which in turn depends on the equipment (magnetic field strength and radiofrequency coil), the pulse sequence used, and post-processing steps such as noiseremoval. Thus, increased resolution could be achieved with higher magnetic field strengths, more sensitive RF coils, more advanced DMI pulse sequences, and improved methods for spectral denoising if available. We have used the best configuration available to us and discussed such limitations in the manuscript, including now a few modifications to address the Reviewer’s point more clearly – page 19 (lines 6-10): “Therefore, further DGE-DMI preclinical studies aimed at detecting and quantifying relatively weak signals, such as tumor glutamateglutamine, and/or increase the nominal spatial resolution to better correlate those metabolic results with histology findings (e.g in the tumor margin), should improve basal SNR with higher magnetic field strengths, more sensitive RF coils, and advanced DMI pulse sequences [55]”.

(3) The authors might consider measuring the contribution of stromal cells and infiltrative immune cells in the analysis of DGE DMI data, to construct a more comprehensive picture of the microenvironment.

RA-R3.C3: Thank you for this important point. We now added additional Iba-1 stainings of infiltrating microglia/macrophages, for each tumor, as suggested by the Reviewer; stromal cells would be more difficult to detect and we did not have access to a validated staining method for doing so. Our new data and results - now included in Fig 2B – indicate significantly higher levels of Iba-1 positive cells in CT2A tumors compared to GL261, which are particularly noticeable in the periphery of CT2A tumors and consistent with their better-defined margins and lower infiltration in the brain parenchyma. This has been explained more extensively in PRR1.1.

(4) Additional GBM models with improved understanding of the genetic markers would serve as an optimal validation cohort to support the potential clinical translation.

RA-R3.C4: We agree with the Reviewer and direct again to RA-R1.3, where we already addressed this suggestion in detail and introduced modifications to the manuscript accordingly.